# Concomitant phytonutrient and transcriptome analysis of mature fruit and leaf tissues of tomato (*Solanum lycopersicum* L. cv. Oregon Spring) grown using organic and conventional fertilizer

**Richard M. Sharpe**[1,2], **Luke Gustafson**[1¤a], **Seanna Hewitt**[1,2], **Benjamin Kilian**[1,2¤b], **James Crabb**[1], **Christopher Hendrickson**[1¤c], **Derick Jiwan**[1,2¤d], **Preston Andrews**[1], **Amit Dhingra**[1,2]*

1 Department of Horticulture, Washington State University, Pullman, WA, United States of America,
2 Molecular Plant Sciences Graduate Program, Washington State University, Pullman, WA, United States of America

¤a Current address: University of Maryland Extension, Charles County, Bel Alton, MD, United States of America
¤b Current address: Liberty University, Lynchburg, VA, United States of America
¤c Current address: Department of Mathematics and Natural Sciences, National University, La Jolla, CA, United States of America
¤d Current address: Hazel Technologies, Chicago, IL, United States of America
* adhingra@wsu.edu

## Abstract

Enhanced levels of antioxidants, phenolic compounds, carotenoids and vitamin C have been reported for several crops grown under organic fertilizer, albeit with yield penalties. As organic agricultural practices continue to grow and find favor it is critical to gain an understanding of the molecular underpinnings of the factors that limit the yields in organically farmed crops. Concomitant phytochemical and transcriptomic analysis was performed on mature fruit and leaf tissues derived from *Solanum lycopersicum* L. 'Oregon Spring' grown under organic and conventional fertilizer conditions to evaluate the following hypotheses. 1. Organic soil fertilizer management results in greater allocation of photosynthetically derived resources to the synthesis of secondary metabolites than to plant growth, and 2. Genes involved in changes in the accumulation of phytonutrients under organic fertilizer regime will exhibit differential expression, and that the growth under different fertilizer treatments will elicit a differential response from the tomato genome. Both these hypotheses were supported, suggesting an adjustment of the metabolic and genomic activity of the plant in response to different fertilizers. Organic fertilizer treatment showed an activation of photoinhibitory processes through differential activation of nitrogen transport and assimilation genes resulting in higher accumulation of phytonutrients. This information can be used to identify alleles for breeding crops that allow for efficient utilization of organic inputs.

**Data Availability Statement:** All relevant data are within the manuscript and its Supporting Information files.

**Funding:** This research was funded in part by CSANR BIOAg grant to PA and AD, and by the USDA National Institute of Food and Agriculture, Hatch project WNP00011 to AD. SLH and BRK acknowledge the support of NIH/NIGMS through institutional training grant award T32-GM008336. The contents of the publication are solely the responsibility of the authors and do not necessarily represent the official views of the NIGMS or NIH. The funders had no role in study design, decision to publish, or preparation of the manuscript.

**Competing interests:** The authors have declared that no competing interests exist.

## Introduction

As agriculture prepares to support 10 billion people by 2050, grow 70% more food in increasingly less amount of land and reduce environmental impact [1], there is a need to assess different modes of farming. The proponents of organic farming have stated that it can have reduced environmental impact from savings in fossil fuels, reduced environmental pollution, and fostering of greater biodiversity however, these benefits accrue at a cost of reduced yield which can range from 5–34% depending on the conditions used [2–4].

A recent meta-analyses, based on data from 343 peer-reviewed publications using a standardized weighted analysis, concluded that on an average, organically grown crops have higher concentrations of antioxidants and a lower percentage of pesticide residues compared to conventional crops across regions and production seasons [5]. This report is in contrast to some previously reported analyses that found no significant difference between the foods grown in conventional vs. organic systems [6, 7]. Several studies have reported the biochemical analyses of organically grown apple, strawberry, and tomato fruits where the levels of antioxidants, phenolic compounds, carotenoids and vitamin C were seen to be enhanced. [8–10]. Based on the incidence of higher phytonutrient content, organically produced foods have also been proposed to be more nutritious compared to their conventionally grown counterparts however, the scientific opinion remains divided [11].

Organic farming necessitates that the nitrogen needs of the plant are met from manures derived from animal byproducts, crop residues, green manures, legumes and soil organic matter [12]. However, the process of developing new crops via plant breeding utilizes conventional agricultural practices, thereby selecting crops with an inherent genetic bias towards utilization of conventional fertilizers and management practices [13]. The fundamental difference between conventionally or organically grown foods is the chemical nature of the nutrition inputs that a plant must utilize. The presence of carbon-linked inorganic nutrients impacts the bioavailability of the inorganic elements. It is assumed that microbial communities first degrade the organic nutrients, which are then absorbed by the plant [14]. Therefore, organic production systems are far more complex than conventional systems and the nutrition of the plant must depend more on the available microbial communities. This begs the following questions: Do the plants, having been bred under conventional inputs, adjust their response at the genetic level to utilize organic nutritional inputs, and interactions with available microbial communities? Is there a natural degree of malleability rendered by some genetic backgrounds to adapt better to the organic inputs? Are there plant metabolic pathways that could be altered with external, but organic, inputs to overcome the yield losses reported under organic conditions?

A very limited number of gene expression studies have attempted to address the aforementioned questions to gain valuable insights into plant's genomic and metabolic performance under organic and conventional fertilizer regimens. A microarray based gene expression study in potato was used to identify statistically different expression profiles under organic and conventional treatments [15]. Further, based on different gene ontology pathways, differentially expressed genes were also identified [16]. In wheat, univariate and multivariate statistical analysis was done on microarray-based gene expression data to establish that the organic inputs influence the expression of global wheat transcriptome, which could be utilized to verify the production system at the farm level [17]. Perhaps, due to the confounding impact of the environment, none of these reports were able to isolate and quantify the impact of organic inputs on a plant's metabolic and transcriptomic response. Furthermore, none of the previous studies reported changes in both the phytochemical composition and the concomitant changes in the global gene expression.

The major concern with organic inputs is the losses in yield which are counter to what is needed to feed a burgeoning population [18]. However, the organic agricultural practices continue to grow and find favor with customers making it both critical and urgent to gain an understanding of the molecular underpinnings of the factors that limit the yield in organically farmed crops. It has been proposed that with proper management practices, crops yields obtained under organic farming can match conventionally farmed crops [19]. Understanding the molecular basis can help in identifying proactive and predictive agronomic, agrochemical or biological interventions that can enhance crop productivity in organic systems. This can have significant implications in enabling small-holding farms across the globe, which have relatively easier access to organic inputs compared to conventional inputs, to grow food sustainably.

In order to understand the impact of conventional and organic inputs on a single genotype of tomato, without the confounding factor of the environment, this study was conducted under controlled environmental conditions. Two hypotheses were evaluated in this study: 1. Organic soil fertilizer management, which results in a slower rate of biological release of available nitrogen to plant roots, results in greater allocation of photosynthetically derived resources to the synthesis of secondary metabolites, such as phenolics and other antioxidants, than to plant growth, and 2. Genes involved in changes in the accumulation of phytonutrients under organic fertilizer regime will exhibit differential expression, and that the growth under different fertilizer treatments will elicit a differential response from the tomato genome. Both these hypotheses were supported, suggesting an adjustment of the plants' metabolic and genomic activity in response to different nitrogen regimes.

## Results and discussion

### Agronomic and phytonutrient analysis

The mean mass of red ripe fruit was 20% greater in conventionally (CONV) grown plants (S1 Fig). Similarly, the cumulative above and below ground vegetative biomass on a fresh weight basis was approximately 4% greater in CONV grown plants. While the above ground vegetative biomass on a dry weight basis was 4.5% higher in CONV grown plants, the below ground vegetative biomass on a dry weight was 6% greater in organically (ORG) grown plants (S1 Table). The observed reduction in yield and biomass under ORG fertilizer is as expected since the use of organic fertilizer has been implicated in the reduction of yield by 5–34% [4].

Total soluble solids were higher by 18.9% in ripe ORG fruit, which also had higher concentrations of phenolic compounds (FW– 17.02%, DW– 16.81%), lycopene (FW and DW– 11.93%), and vitamin C (FW– 11.46%, DW– 13.88%) compared to CONV fruits (Table 1). Leaf C: N ratio was higher by 12.5% in ORG plants, thus most likely favoring the synthesis of C-based compounds, like phenolics, ascorbic acid, and carotenoids (S2 Table). Similar results regarding accumulation of phytonutrients has been reported in several previous studies [9, 10]. The observation of enhanced levels of phytonutrients supported the first hypothesis that growth of tomato plants under organic fertilizer management results in greater accumulation of secondary metabolites, such as phenolics and other antioxidants, than to plant growth. This may be due to the preferential allocation of photosynthetically derived resources to the synthesis of these secondary metabolites.

### RNAseq and biochemical pathway analysis

**Transcriptome summary.**  A tabulation of total raw sequence reads, number of curated reads, number of assembled contiguous sequences (contigs) or expressed sequence tags (ESTs), and number of ESTs exhibiting very high differential expression ratios, equal to or

**Table 1. Measured phytochemical concentration of red ripe tomato fruit.** Soluble solids, total phenolics (gallic acid equivalents, GAE), lycopene, total Trolox equivalent antioxidant capacity (TTEAC), lypophilic TEAC, hydrophilic TEAC, and reduced ascorbic acid concentrations on fresh weight (FW) and dry weight (DW) bases (except soluble solids) under conventional (CONV) and organic (ORG) fertilizer treatments. Data were analyzed using SAS Mixed Model. (See Supplementary file for measured observations, LS means, standard deviation, and standard error).

| Variable | Units | Main Effect Means | | % change from CONV |
|---|---|---|---|---|
| | | CONV | ORG | |
| Soluble Solids | °Brix | 5.04 | 6.03** | 19.4 |
| Total Phenolics | mg GAE/g FW | 0.235 | 0.275*** | 17.0 |
| | mg GAE/g DW | 22.0 | 25.7*** | 16.8 |
| Lycopene | mg/g FW | 3.77 | 4.22** | 11.9 |
| | mg/g DW | 352 | 394** | 11.9 |
| Total TEAC | mmol/g FW | 2.88 | 3.07** | 6.59 |
| | mmol/g DW | 269 | 286** | 6.31 |
| Lipophilic TEAC | mmol/g FW | 1.48 | 1.61** | 8.78 |
| | mmol/g DW | 138 | 151** | 9.42 |
| Hydrophilic TEAC[z] | mmol/g FW | 1.27 | 1.45*** | 14.2 |
| | mmol/g DW | 119 | 135*** | 13.8 |
| LTEAC:TTEAC | ratio | 0.51 | 0.52[NS] | 1.96 |
| HTEAC:TTEAC | ratio | 0.45 | 0.48* | 6.67 |
| Ascorbic acid, reduced | mg/g FW | 157 | 175*** | 11.5 |
| | mg/g DW | 14.3 | 16.3** | 13.9 |

*$p \leq 0.05$

**$p \leq 0.01$

***$p \leq 0.001$

[NS]non-significant

[z]Failed to meet normality, based on Kolmogorov-Smirnov test.

greater than 5-fold expression difference, between the ORG and CONV treatments for each tissue and treatment is summarized in S3 Table. A total of 2,324 and 3,035 contigs were identified to exhibit equal to or greater than 5-fold expression difference between the ORG and CONV treatments in fruit and leaf samples, respectively.

**Quantitative RT-PCR verification.** The expression trends obtained from the RPKM values were verified by performing RT-qPCR based expression of 27 selected genes. Of these, eight genes were selected to be evaluated as reference genes based upon the fact that their tissue and sample specific RPKM ratios were nearly zero, which is indicative of equivalent expression between the tissue and sample types. From these eight candidate genes, Calreticulin3-like, accession number Solyc05g056230.3.1, was selected as the reference gene due to the equivalent tissue specific expression values obtained from RPKM and RT-qPCR results (S2 Fig).

**Functional annotation.** The transcriptome data was processed using Blast2GO sequence alignment, gene ontology (GO) mapping, and functional annotation workflow [20–22]. Sequences were processed through BLAST against the Viridiplantae database using an e-value cutoff of 1.0e−3. After Blast2GO processing, the relative expression of contigs (RPKM values) with GO terms involved in the differential accumulation of phytonutrients was analyzed.

## Global changes in gene expression under CONV and ORG conditions

Observed differences in the phytonutrient and agronomic characteristics of tomato plants grown under ORG and CONV fertilizer treatments indicated growth under different fertilizer

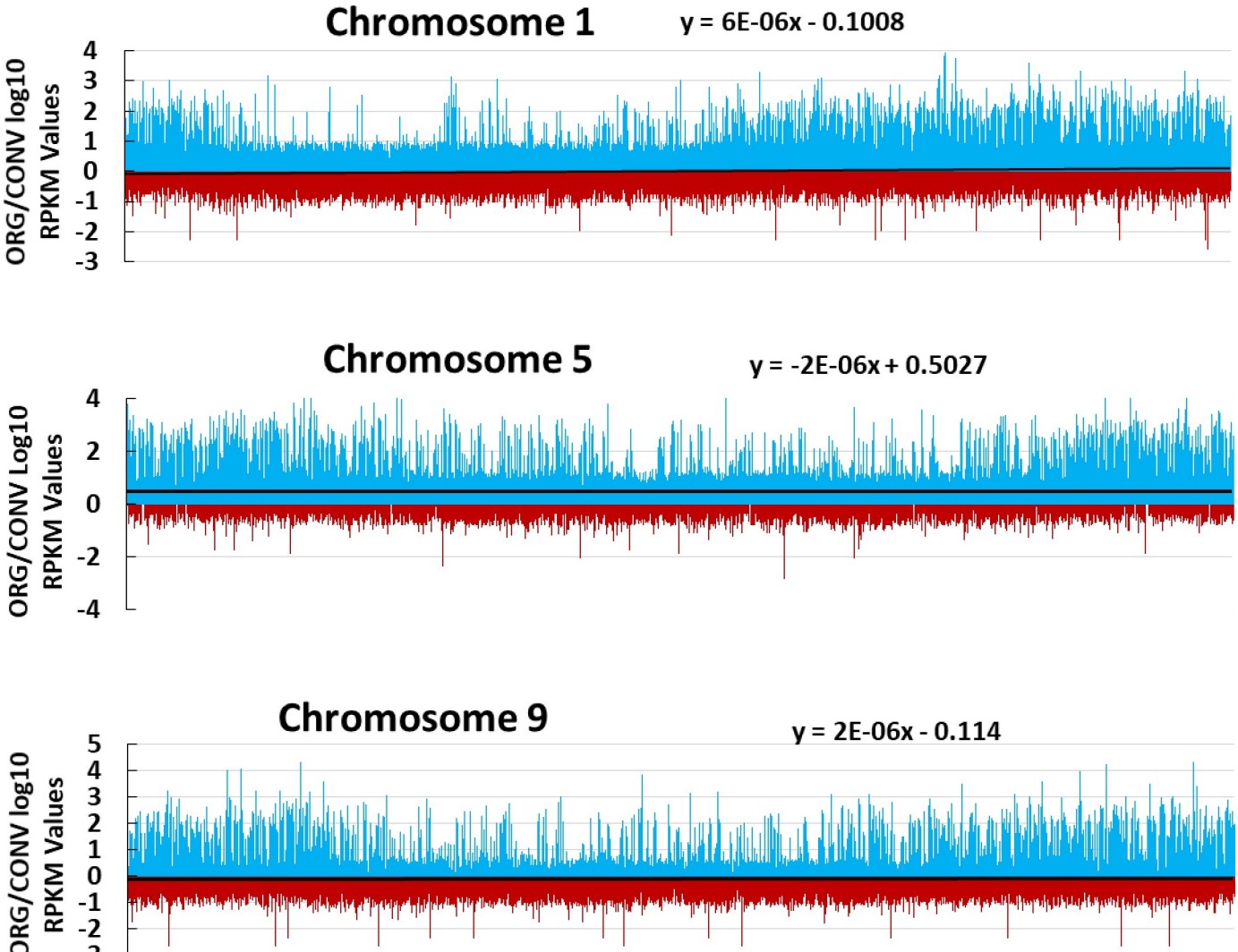

**Fig 1. Representative ORG and CONV log10 RPKM ratio Manhattan plot representations of expression values across *S. lycopersicum* chromosomes.** Every bar is indicative of the log10 expression ratio of Fruit and Leaf ORG RPKM values over Fruit and Leaf CONV RPKM values. Trendlines with slope equations indicate propensity of treatment activated loci across the chromosome. Chromosome graph lengths are not indicative of chromosome length. Red bars indicate a higher log10 ratio in the CONV treatment and blue bars indicate higher expression in the ORG treatment at any given position on the chromosome. Trend lines were graphed to indicate a predominance of chromosomal activity either in the ORG treatment (positive y-intercept value) or in the CONV treatment (negative y-intercept value). All 12 mapped chromosomes can be found in S3 Fig. 12 read files, representing three cDNA libraries across each of the two tissue and two treatment types were used for mapping purposes.

treatments is expected to elicit a differential expression response at a global level. Visualization of the mapped $log_{10}$ ratio of RPKM values (ORG/CONV) across each chromosome of tomato reflected the difference in expression. The observed shift of the trend lines, indicative of a predominance of overall chromosomal expression either in the ORG treatment (positive y-intercept value) or in the CONV treatment (negative y-intercept value), indicated an enhanced expression at certain genomic loci in ORG grown plants (Fig 1 and S3A and S3B Fig). This observation supports a part of the second hypothesis that growth of the same genotype of tomato under different fertilizer treatments will elicit a differential expression response from the tomato genome.

## Gene Ontology and KEGG Pathway analysis related to Phytonutrient biosynthesis and Primary Metabolic pathways

Several recent studies have utilized transcriptomics-based gene ontology and Kyoto Encyclopedia of Genes and Genomes (KEGG) pathway analysis for establishing a theoretical basis to understand the metabolic framework underlying complex developmental processes or organismal responses to external cues in tomato and other species [23–28].

In this study, GO and KEGG pathway analysis of the transcriptome data focused on genes related to biosynthesis of phytonutrients that were observed to accumulate differentially under the two conditions. Prior studies have attributed the observed changes in metabolites primarily to stress, however, the role of nitrogen source was left unaddressed [29, 30]. Besides the organic forms of nitrogen, organic fertilizer predominantly contain amino acids and ammonia, which is known to be toxic to tomato [31]. Efforts to overcome the toxicity of ammonia likely activates the salvage pathways that include photorespiratory processes and production of phytonutrients protective to the plant. Since ammonia impacts nitrogen metabolism and photosynthetic processes in the plant [32], the biochemical pathway analysis of the transcriptome data also included an evaluation of the two primary metabolic pathways to gain an understanding of their impact and interaction when the plants were grown under the two fertilizer conditions.

## Impact of ORG inputs on Lycopene and Ascorbate pathways

**Lycopene.** An approximate increase of 12% in Lycopene content was observed in the fruit, both on a fresh and dry weight basis under ORG fertilizer (Table 1). Analysis of the expression values of all the genes involved in the Lycopene pathway revealed that Phytoene synthase 1 Solyc03g031860.3.1, representing the first committed enzyme in the lycopene pathway, was the highest expressed transcript in the pathway in the fruit tissue compared to the leaf tissue, however there were no differences in the fruit transcripts from the two treatments. Interestingly, the expression of Phytoene synthase 1 (*Psy1*) was significantly higher, log2 Fold Change (log2FC) 2.225, in the ORG leaf tissue as well as 15-cis-zeta-carotene isomerase (Z-ISO) Solyc12g098710.2.1 with a log2FC value of 1.949 (Fig 2A & 2D). Phytoene synthase 2 (*Psy2*) (Solyc02g081330.4.1), a chloroplast targeted specific homolog exhibited an overall low expression in both tissues and did not indicate a difference between the treatment types (S4 Fig).

The first question that emerges from these observations is what is contributing to the higher levels of lycopene in the fruit? Capture of the fruit transcriptome expression at the mature stage does not provide any relevant information. Perhaps, the higher accumulation could be attributed to higher activity of Psy1 or other enzymes involved in the pathways during breaker or earlier stages of fruit development. The second question is why the levels of *Psy1* and *Z-ISO* are higher in ORG leaves and what function may they be serving? Psy1 is active primarily in chromoplasts, and Psy2 is a chloroplast predominant enzyme [33]. The lycopene pool fluxes through lycopene β-cyclase, in the lycopene B pathway producing β-carotene, antheraxanthin, violaxanthin and neoxanthin, or through lycopene ε-cyclase in the lycopene A pathway producing α-carotene and lutein [33]. Upregulation of *Psy1* and *Z-ISO* in the leaf under ORG conditions is intriguing. One explanation could be that the additional activity of *Psy1* is resulting in the production of additional pools of lycopene that may be converted into carotenoid or abscisic acid (ABA) via activation of 9-cis-epoxycarotenoid dioxygenase (NCED) converting violaxanthin to xanthoxin to compensate for photorespiratory stress [33]. Lycopene cyclases, enzymes responsible for converting lycopene metabolite pools for downstream carotenoid production, were down regulated in the fruit tissue compared to the leaf tissue, which, as

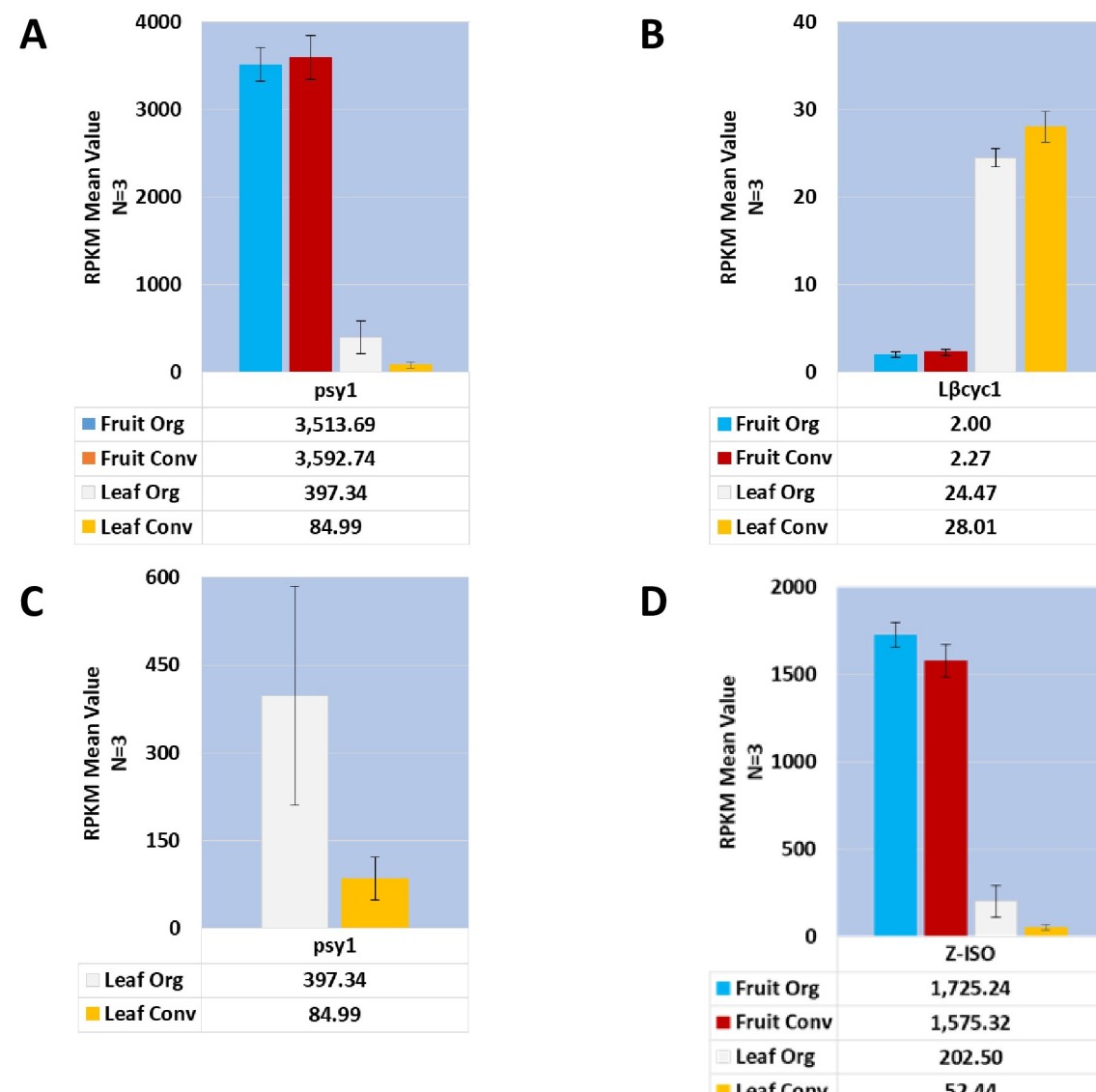

| Gene Abbreviation | SOL Accession | Annotation | Lab Annotation | Org/CONV Fruit log2FC | ORG/CONV Leaf log2FC |
|---|---|---|---|---|---|
| psy1 | Solyc03g031860.3.1 | Phytoene synthase 1 | contig 673 | -0.032099617 | 2.224928329 |
| Lβcyc1 | Solyc04g040190.1.1 | lycopene beta-cyclase | contig 21075 | -0.185228959 | -0.195248596 |
| Z-ISO | Solyc12g098710.2.1 | 15-cis-zeta-carotene isomerase | contig 205 | 0.131152718 | 1.949135928 |

**Fig 2. Mean expression profile of the Lycopene pathway committed steps and significant differential enzymes.** PSY1 is the rate limiting step in the carotenoid biosynthesis pathway and the absence of LβCYC in the pathway increases the pool of lycopene from the pathway. **A.** *psy1* RPKM expression values. **B.** *Lβcyc1* RPKM expression values. **C.** *psy1* leaf expression values with a different *y*-axis to discern standard error differences. **D.** *Z-ISO* leaf expression values. Table of annotations and accession numbers from Sol Genomics ITAG3.2. Error bars denote the standard error of sequenced expression values between 3 biological replicates.

expected is responsible for higher lycopene content in the fruit tissue compared to leaf (S4 Fig). Increased lycopene cyclase activity, as well as between the beta and epsilon isoforms, in tomato fruit has been shown to decrease lycopene content [34, 35]. Absence of differential activity of *Psy1* and the lycopene cyclases in the ORG and CONV fruit tissues indicates that there is a need for a developmental time course transcriptome analysis under these two conditions to identify the molecular reason underlying higher observed lycopene content in ORG fruit.

**Biosynthesis and recycling of ascorbate.** In tomato the major pathway leading to accumulation of Ascorbic acid (AsA) is the Smirnoff-Wheeler pathway [36], whereas the recycling of AsA to control oxidative stress is regulated by the Foyer-Halliwell-Asada pathway [37]. Transcripts representing the enzymes involved in biosynthesis and recycling of AsA were assessed for their expression levels.

**Biosynthesis of AsA—Smirnoff-Wheeler pathway.** Depending on the plant species, transcriptional activity of two genes has been reported to be directly implicated in the overall production of AsA [38]. In Arabidopsis and kiwi fruit, GDP-L-galactose phosphorylase/guanyltransferase (GGP) was identified as a rate limiting step for synthesis of AsA [39]. In ripening tomatoes, the overall production of AsA was reported to be dependent on the expression of L-galactose-1-phosphate phosphatase (GPP) [40]. However, overexpression of GGP resulted in a 3- to 6-fold increase of AsA in transgenic tomato fruit indicating GGP expression can also contribute to total AsA in tomato fruit [41]. In this study, transcriptional abundance of *GGP1* Solyc06g073320.3.1 had a log2FC of -0.398 between the fruit tissue and a log2FC value of -0.363 between leaf tissues indicating they were slightly down regulated in the ORG tissues. Expression profiles for the two isoforms of GPP, *GPP1* and *GPP2*, indicate *GPP1* was expressed in a fruit tissue manner and *GPP2* in both leaf and fruit manner although at diminished and non-differential levels. The expression of the L-galactono 1,4-lactone dehydrogenase (*galLDH*) Solyc10g079470.3.1 gene was slightly down regulated in the fruit tissue with a log2FC value of -0.385 (Fig 3, S5 Fig, S4 Table). GDP-D-mannose 3,5-epimerase expression identified the presence of two isoforms, *GME1* and *GME2*, with a log2FC value of -0.627 for the *GME1* Solyc09g082990.3.1 isoform in the fruit tissue indicating a moderate down regulation in the ORG fruit tissue. Mannose-6-phosphate isomerase (*MPI*) was represented by two different isoforms where *MPI2* Solyc02g063220.3.1 was moderately down regulated in the ORG treatment as indicated by a log2FC value of -0.537 in the leaf tissue.

**Recycling of AsA—Foyer-Halliwell-Asada pathway.** The metabolic functions of ascorbate in plant tissue require a recycling pathway referred to as the Foyer-Halliwell-Asada pathway [42]. The expression of genes coding for the Foyer-Halliwell-Asada enzymes was evaluated (Fig 3, S6 Fig, S5 Table).

As a part of the reduction/oxidation of glutathione branch of the pathway [42], two isoforms of glutathione reductase (*GSR1* and *GSR2*) (Solyc09g091840.3.1 and Solyc09g065900.3.1, respectively) and one isoform of dehydroascorbate reductase (*DHAR*) (Solyc05g054760.3.1) were identified. *GSR1* was moderately down regulated in the ORG leaf tissue, log2FC -0.500, (Fig 3). The *GSR2* isoform was expressed at a lower level compared to *GSR1* with no discernable differences between the ORG and CONV treatments. *DHAR* was slightly upregulated in the ORG fruit tissue, log2FC 0.319, and slightly down regulated in the ORG leaf tissue having a log2FC value of -0.407.

Interestingly, evaluation of the expression levels of the enzymes responsible for the cycling of redox status of glutathione in the Foyer-Halliwell-Asada cycle indicate that the greatest transcript activity occurs in the leaf tissue. Enzymes responsible for the detoxification of $H_2O_2$—members of the ascorbate peroxidase and superoxide dismutase families—were expressed at a higher level in the fruit tissue, except for one cytosolic ascorbate peroxidase

# Ascorbate related genes

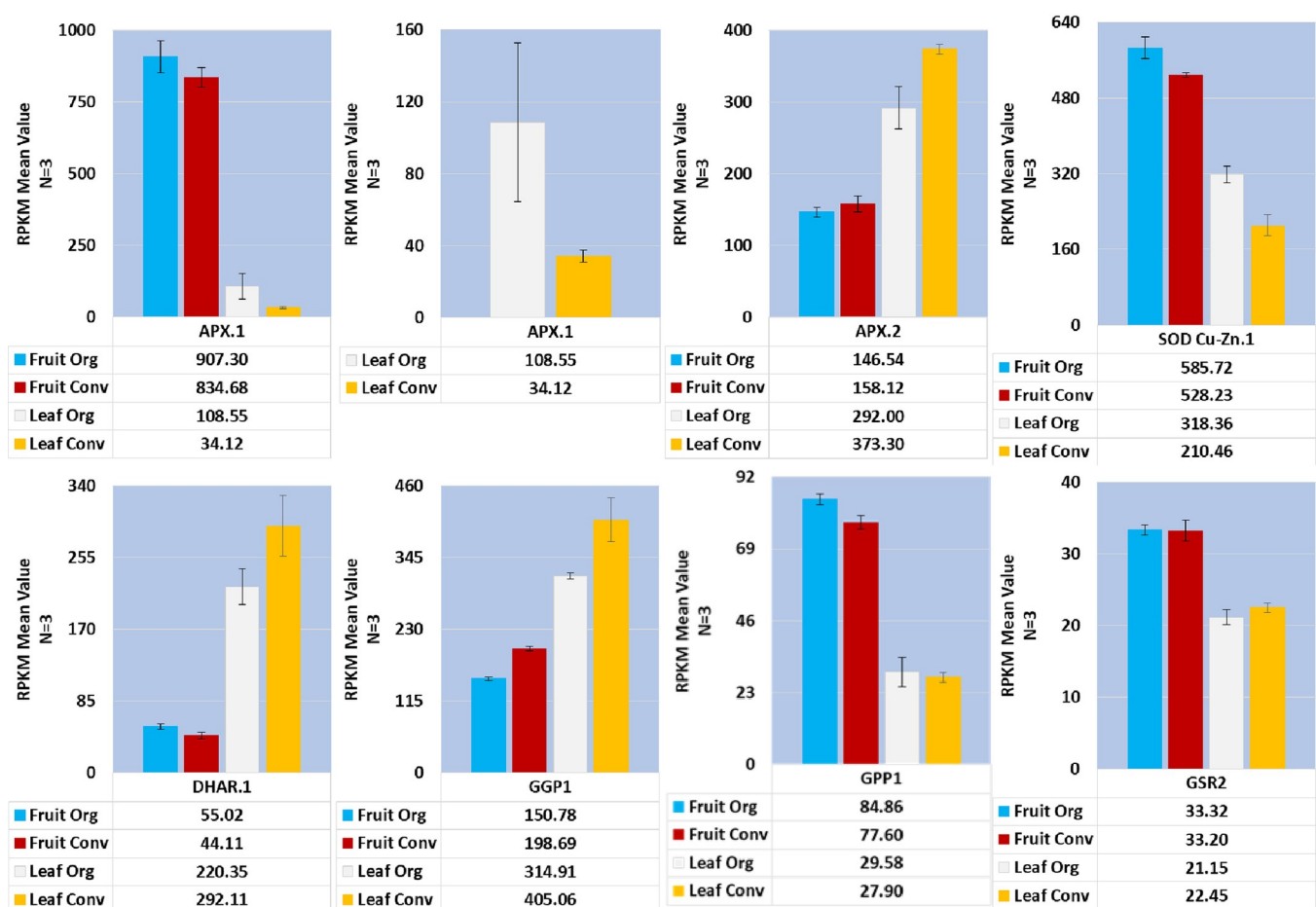

| Abbreviation | Sol Accession | Annotation | Lab Annotation | Org/CONV Fruit log2FC | ORG/CONV Leaf log2FC |
|---|---|---|---|---|---|
| APX.1 | Solyc09g007270.3.1 | L-ascorbate peroxidase cytosolic | Contig 1593 | 0.12034786 | 1.669611113 |
| APX.2 | Solyc06g005160.3.1 | L-ascorbate peroxidase cytosolic | Contig 188 | -0.109694812 | -0.354385168 |
| SOD Cu-Zn.1 | Solyc01g067740.3.1 | superoxide dismutase [Cu-Zn] 1 | Contig 494 | 0.149040748 | 0.597113359 |
| GSR2 | Solyc09g065900.3.1 | glutathione chloroplastic | Contig 2130 | 0.005066154 | -0.086236913 |
| DHAR.1 | Solyc05g054760.3.1 | dehydroascorbate reductase | Contig 1336 | 0.31869902 | -0.406740178 |
| GGP1 | Solyc06g073320.3.1 | GDP-L-galactose phosphorylase 2-like | Contig 1951 | -0.398072293 | -0.363187854 |
| GMP1 | Solyc06g051270.3.1 | mannose-1-phosphate guanylyltransferase 1-like | Contig 2843 | -0.090495318 | 0.447484856 |

**Fig 3. Mean expression profile of significant ascorbate pathway enzymes.** GGP1 and GMP1 are involved in the production of ascorbate (Smirnoff-Wheeler Pathway). APX.1, APX.2, SOD Cu-Zn.1, GSR2 and DHAR.1 are involved in the activation of ascorbate (Foyer-Halliwell-Asada Pathway). Table of annotations and accession numbers from Sol Genomics ITAG3.2. Additional ascorbate pathway gene expression can be found in S5 and S6 Figs. Error bars denote the standard error of sequenced expression values between 3 biological replicates.

(*APX2*) (Solyc06g005160.3.1) transcript, and a peroxisomal APX (*peroxAPX*) (Solyc01g111510.3.1). (Fig 3). Four additional *APX* isoforms were identified with *APX1* (Solyc09g007270.3.1) being significantly upregulated, log2FC 1.670, in the ORG leaf tissue and *APX2* slightly down regulated, log2FC -0.354, in the ORG leaf tissue (S6 Fig).

A cytosolic superoxide dismutase (*SOD*) transcript belonging to the copper-zinc (Cu-Zn) family (Solyc01g067740.3.1) demonstrated a moderate increase in ORG treatment transcription, log2FC of 0.597, in the leaf tissue (Fig 3). Additional *SOD* transcripts were also detected; one *SOD Cu-Zn*, a chloroplast targeted SOD Cu-Zn (*cpSOD Cu-ZN*), four chloroplast targeted SOD Fe family isoforms and a mitochondria targeted SOD Mn family isoform (S6 Fig and S5 Table). Notably, the *cpSOD Cu-Zn* transcript expression values were higher in the fruit tissue but significantly higher in the ORG leaf tissue (log2FC 1.648).

An obvious correlation between the observed levels of ascorbate in the fruit tissue with the expression patterns of genes coding for enzymes in the ascorbate biosynthesis and recycling of ascorbate is missing. This is most likely due to the fact that in this study mature fruit tissues were used for the transcriptome analysis necessitating a time course developmental biochemical and transcriptome analysis of the fruit tissue to understand the role of all the genes involved in ascorbate biosynthesis. Higher overall expression of transcripts coding for enzymes involved in recycling in the ORG leaf tissues is interesting. It indicates that ascorbate recycling may be operating at a higher level under ORG conditions and may contribute to higher ascorbate levels in the fruit. Although fruit microclimate has been shown to influence fruit ascorbate levels as well [43, 44].

**Impact of ORG inputs on biosynthesis of TEAC, total phenolics, and soluble solids.**
Three of the five phytonutrients quantified in this study, namely TEAC, total phenolics, and soluble solids, represent a combination of metabolic components rendering it difficult to ascribe the contribution of specific enzymes towards the observed final concentration. For instance, the TEAC test for antioxidant capacity does not specifically test for individual antioxidants, but rather total antioxidant capacity, and should not be used to identify individual antioxidants by kinetics or stoichiometry [45]. As discrete antioxidant activity cannot be ascertained with this method, it was not feasible to analyze the expression of genes coding of enzymes involved in these pathways. Similar to the TEAC test, the Folin-Ciocalteu assay for total phenolic content assessment [46], reacts with at least 33 different phenols or compounds when the assay is performed on total tissue eluates [46, 47]. Therefore, the assay could not be used to quantify levels of any particular phenol without specific metabolite assays. Sugars, organic acids, amino acids and soluble pectins constitute the majority of soluble solids reported as °Brix values from refractometer measurements. Given the complexity of the metabolites being detected for the three classes of phytonutrients, it would not be informative to draw any correlations to specific genes that participate in the biosynthetic pathways of these phytonutrients. Therefore, the Gene Ontology (GO) Fisher's exact test was utilized as an alternative approach to identify the differential expression of genes involved in these pathways.

Comparison between the combined Leaf ORG and CONV treatments (leaf subset) and the combined reference yielded no significant GO term enrichment. Since there was no enrichment of GO terms in the leaf tissue, it should not be assumed that the response of individual transcripts in the leaves were similar under the two conditions. This is illustrated by the

RPKM values of *Psy1*, which were 2.2-fold higher in the ORG treatment versus the CONV treatment (Fig 2).

However, interestingly, among the combined fruit ORG and CONV (fruit subset) comparison, terms associated with the chloroplast, photosynthesis, organic acid cycling processes and functions of the plastids, specifically the chloroplast, thylakoid and photosystems, and sulfur compound biosynthesis were over represented in the Fruit ORG tissues (Fig 4). In the Fruit ORG treatment there was a higher representation of genes related to photosynthesis and photosynthetic stress. The underrepresented fruit ORG GO terms included non-specific organelle, structural molecule activity and translational machinery (Fig 4). The underrepresented GO terms in the ORG fruit were involved in the cellular component ontology, mainly involved in cell structure components. Underrepresented GO terms in the three ontology categories represented greater than half of the contigs that showed a lower expression of 2-fold or lower values with the exception of three GO terms.

When the contigs corresponding to the enriched GO terms were mined and their RPKM values compared, it was revealed that all of them demonstrated a 2-fold greater RPKM value for the ORG treatment than either the CONV or non-differential RPKM values (S7 Fig). One exception was GO: 0043226 organelle cellular component, which had a larger proportion of contigs that were non-differentially expressed (S7 Fig). While the number of contigs expressed at 2-fold or higher level in the ORG treatment for two of the terms, GO:0005622 and GO:0005634 comprised 50% of the contigs, one term, GO:0043226 was comprised of more non-differentially expressed contigs with 34.6% of the contigs expressed at 2-fold or higher level in the ORG treatment (S7 Fig).

GO term enrichment between the differentially expressed leaf and fruit ORG subset and the leaf and fruit CONV subset also returned greater ratio of contigs represented in the ORG subset. Surprisingly, 10 of the 23 ontology terms with higher ORG ratios included under-represented GO terms, i.e. GO:0005622, GO:006412 and GO:0005198, and the Cellular Component ontology had two times more underrepresented GO terms than overrepresented GO terms with 8 and 4 respectively, suggesting more transcriptional activity was involved in structural components of the plant cells in the CONV treatment (S7 Fig).

With the overall transcript activity enrichment partitioned between structural components in the CONV treatment and the photosynthetic and organic acid cycling terms in the ORG treatment, the contigs with differential expression between tissue specific transcripts under the two treatment regimens were extracted. This was done to sort and identify the transcripts with the highest expression differences with a generalized linear model (GLM) and select the associated upper quartile slice for the GO terms (S8 Fig).

Differential expression enrichment based on GO terms in the CONV fruit tissue indicated higher representation of transcripts involved in RNA processing and oxidoreductive processes. The same four GO terms (S8 Fig) were enriched in both the GLM, and the upper quartile model. In the leaf tissue, both the GLM and upper quartile model comprised of overrepresentation of enriched GO terms for Leaf ORG and underrepresentation of non-differentially expressed contigs in the leaf CONV dataset (S9 Fig). Nine GO terms, with five of the GO terms overrepresented in the ORG leaf tissue, were enriched in the GLM model (S9A Fig) while 13 GO terms, with six of the terms over-represented in the ORG leaf tissue, were included in the upper quartile model (S9B Fig).

The GO terms overrepresented in the differentially expressed categories in the leaf tissues are involved in oxidoreductive and oxalate metabolic processes while the differentially expressed under-represented GO terms are involved with binding of the cyclic compounds of monosaccharides and nucleosides. Categorically, the enriched gene ontology and differential gene expression of the leaf tissue indicates more transcriptional resources are being used for

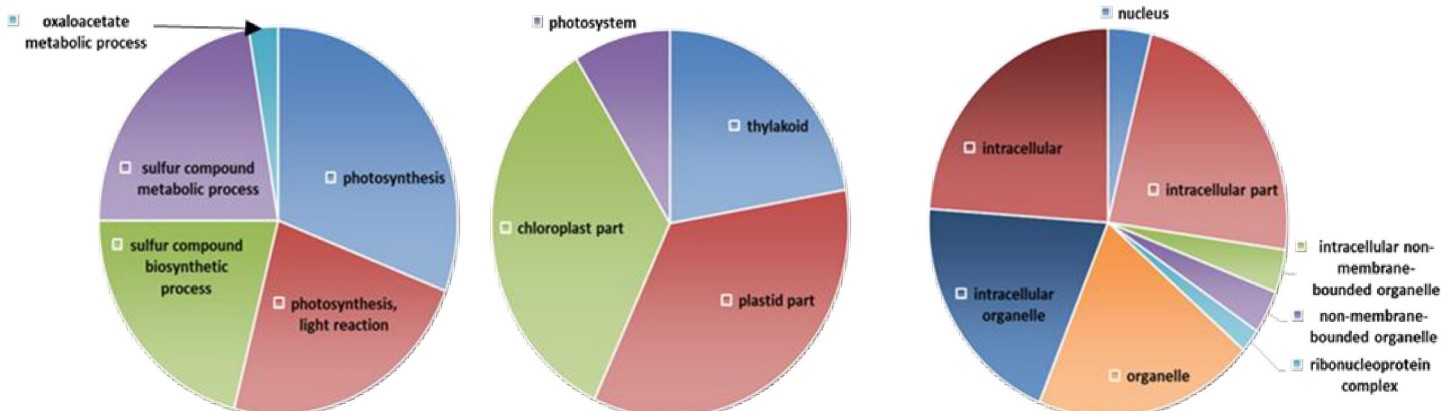

A. Over Represented Biological Process  B. Over Represented Cellular Component  C. Under Represented Cellular Component

| GO-ID | Term | Category | FDR | P-Value | #Test | #Ref | Over/Under |
|---|---|---|---|---|---|---|---|
| GO:0005634 | nucleus | C | 0.010836 | 7.18E-06 | 38 | 560 | under |
| GO:0009579 | thylakoid | C | 0.02231 | 3.33E-05 | 48 | 184 | over |
| GO:0044424 | intracellular part | C | 0.034259 | 7.44E-05 | 228 | 2117 | under |
| GO:0044435 | plastid part | C | 0.034259 | 8.15E-05 | 75 | 348 | over |
| GO:0043232 | intracellular non-membrane-bounded organelle | C | 0.034259 | 8.52E-05 | 35 | 491 | under |
| GO:0043228 | non-membrane-bounded organelle | C | 0.034259 | 8.52E-05 | 35 | 491 | under |
| GO:0044434 | chloroplast part | C | 0.036758 | 0.000102 | 74 | 343 | over |
| GO:0030529 | ribonucleoprotein complex | C | 0.036758 | 0.000104 | 19 | 326 | under |
| GO:0043226 | organelle | C | 0.042013 | 0.000125 | 194 | 1839 | under |
| GO:0043229 | intracellular organelle | C | 0.044346 | 0.000149 | 194 | 1838 | under |
| GO:0005622 | intracellular | C | 0.044346 | 0.000159 | 233 | 2138 | under |
| GO:0009521 | photosystem | C | 0.044346 | 0.000162 | 19 | 49 | over |
|  |  |  |  |  |  |  |  |
| GO:0004611 | phosphoenolpyruvate carboxykinase activity | F | 0.000738 | 2.45E-07 | 8 | 1 | over |
| GO:0008964 | phosphoenolpyruvate carboxylase activity | F | 0.000738 | 2.45E-07 | 8 | 1 | over |
| GO:0005198 | structural molecule activity | F | 0.012726 | 1.22E-05 | 15 | 308 | under |
| GO:0008948 | oxaloacetate decarboxylase activity | F | 0.012726 | 1.46E-05 | 6 | 1 | over |
| GO:0016830 | carbon-carbon lyase activity | F | 0.013502 | 1.79E-05 | 18 | 36 | over |
|  |  |  |  |  |  |  |  |
| GO:0015979 | photosynthesis | P | 0.002779 | 1.38E-06 | 47 | 155 | over |
| GO:0019684 | photosynthesis, light reaction | P | 0.012726 | 1.48E-05 | 35 | 110 | over |
| GO:0044272 | sulfur compound biosynthetic process | P | 0.027498 | 4.56E-05 | 32 | 102 | over |
| GO:0006790 | sulfur compound metabolic process | P | 0.029199 | 5.32E-05 | 34 | 112 | over |
| GO:0006412 | translation | P | 0.044188 | 0.000139 | 22 | 356 | under |
| GO:0006107 | oxaloacetate metabolic process | P | 0.046254 | 0.000176 | 4 | 0 | over |

**Fig 4. Gene Ontology Fisher's Blast2GO analysis and queried with fruit tissues. A.** Biological Process gene ontology terms over represented in the ORG treatment compared to CONV treatment. **B**. Over represented Cellular Components in the ORG treatment versus the CONV treatment. **C.** Cellular Component ontology terms under represented in the ORG treatment compared to the CONV treatment. Size of slices represent ratio of numbers of contigs to total number of contigs in the represented categories.

photosynthetic and energetic transcript activity in the leaf CONV tissue while transcriptional resource allocation in the leaf ORG tissue was targeted towards redox maintenance activity. These observations based on GO enrichment imply that in the absence of the plant needing to divert its resources towards photorespiratory processes in CONV tissue, the metabolism is shifted towards plant growth and development, which supports part of the first hypothesis.

## Phosphate utilization

Inorganic phosphate (P*i*) is a critical element required for plant nutrition especially for photo-synthetic and respiratory metabolism. Several genes implicated in P*i* uptake are under transcriptional regulation under P*i* starvation [48]. Three of these genes; SIZ1, a SUMO E3 ligase, PHO1, a phosphate transporter and SEC12, an endoplasmic reticulum located trafficking protein, have been characterized in *Arabidopsis* as becoming upregulated during P*i* deficiency [49–54]. log2FC expression values of *SIZ1* isoforms indicated no discernable differential expression. *PHO1* isoforms were expressed at very low levels, below an RPKM value of 10. *PHO1x2* was the only member significantly differentially expressed. *PHO1x2* was downregulated in the ORG leaf treatment with a log2FC value of -1.248 indicating greater access to P*i* than in the CONV treatment. *PHO1* was slightly upregulated, log2FG 0.300, in the ORG fruit tissue. *SEC12.1* had a slight upregulation, log2FC 0.302, between leaf treatments. ORG and CONV treatments were similar in their phosphate uptake profiles and any phenotypic as well as metabolite differences would not be due to phosphate deficiencies (S10 Fig).

## Impact of ORG inputs on primary metabolic pathways

The regulation of nitrogen and carbon metabolism in plants is tightly linked. Any perturbations in nitrogen metabolism have a direct impact on the rates of photosynthesis, photorespiration, and respiration [55]. The primary source of nitrogen uptake is through the roots but the terminal sinks for nitrogen, primarily the leaf and fruit of a plant, require nitrogen influx for various metabolic processes. Since the organic fertilizer differs significantly in the type of available nitrogen, different nitrogen uptake pathways were queried for differential expression. For the analysis of this pathway, RT-qPCR based quantitative expression analysis was also included for some of the genes to complement RNAseq data and gain an understanding of how the transporters and enzymes involved in nitrogen metabolism were impacted.

Nitrate studies, as well as other forms of nitrogen assimilation studies, have shown nitrate, nitrite and ammonium elicits transcriptome-specific and regulatory responses in diverse and interconnected ways [56, 57]. Leaf transcript profiles from this study of the nitrate uptake and assimilation components were found to be in general agreement with the induction of transcripts from the introduction of nitrate in CONV fertilizer.

**Nitrate transporters.**   Two major classes of nitrate transporters have been described in the literature. High affinity transporters operate in the 10 to 250μM nitrate concentration range and low affinity transporters operate above the 250μM concentration limit (reviewed in [58]). Four of the nitrate transporters, belonging to the NRT1-PTR family, identified in the ORG vs CONV dataset (Fig 5), have previously been characterized as low affinity nitrate transporters [59]. Of the low nitrate affinity transporters expressed *NRT1PTR2-11* and *NRT1-2-13*, with a log2FC value of -0.447 and -0.435, were slightly down regulated in the ORG leaf tissue. *NRT1-4-6*, log2FC -0.603, was moderately down regulated in the leaf ORG tissue. Two high affinity transporters, *NRT2-1.1* and *NRT2-1.2*, were differentially expressed between the tissue types with *NRT2-1.1* moderately down regulated, log2FC -0.509, in the ORG leaf tissue (Fig 5D & 5E). Nitrate deficiency in Arabidopsis has been shown to result in the up regulation of high affinity nitrate transporters [60] and does not indicate a deficiency of nitrate translocation in the leaves of the ORG fertilizer treatment.

**Nitrite transporters.**   Characterization of nitrite transporters remains elusive in plants as indicated by the lack of previously published literature. Given this constraint, the nitrite transporter sequences from *Arabidopsis thaliana*, *Cucumis sativus* and *Vitis vinifera* were used to query the *S. lycopersicum* NCBI sequence database. The XM_004240292.2 PREDICTED: *Solanum lycopersicum* protein NRT1/ PTR FAMILY 3.1-like orthologous sequence had the highest

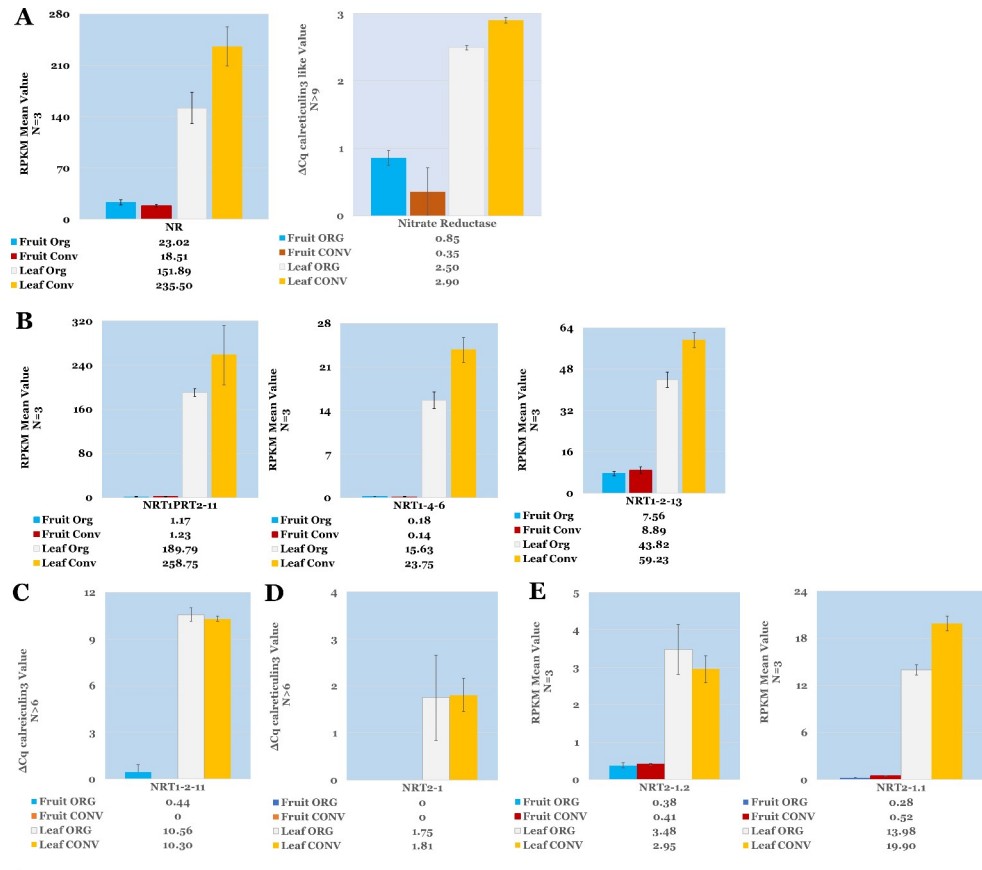

**Fig 5. Expression profile of mean RPKM values of Nitrate Assimilation components in Fruit and Leaf tissue. A.** RPKM and RT-qPCR values for Nitrate Reductase. **B. L**ow affinity nitrate transporters NRT1-2-11 and NRT1-4-6 **C & D.** RT-qPCR expression values for the highest expressed low and high affinity nitrate transporters. **E**. High affinity nitrate transporters NRT2-1.2, NRT2-1.1. RT-qPCR results (A, C and E) are averages of the number of replicates (N) and bars indicate the standard deviation between replicates. Table of annotations and accession numbers from Sol Genomics ITAG3.2. RT-qPCR relative values for tissue type and treatment, trend with the corresponding RPKM values generated from the HTS analysis. Error bars denote the standard deviation of RT-qPCR expression values between 3 biological replicates and at least 3 technical replicates. Error bars denote the standard error of sequenced expression values between 3 biological replicates.

homology to the *V. vinifera* KF649633 and *C. sativus* NM_001280623 NITR1 nitrite transporters. The expression of *S. lycopersicum NITR2* orthologues were barely detectable in this study in the leaf and fruit tissues regardless of fertilizer treatment, and the expression of *S. lycopersicum* NRT1/ PTR FAMILY 3.1-like XM_004240292.2 sequence was not detected at all in the fruit tissue. The Solyc03g113280.3.1 Integral membrane HPP family protein sequence aligned with highest homology to both the Arabidopsis NITR2.1 BT030052 and NITR2.2 BT006407

nitrite transporters and was shown to be slightly down regulated, log2FC -0.417, in the ORG leaf tissue (Fig 6C). Thus, no conclusion could be drawn regarding the role of nitrite transporters in this experimental set up.

**Ammonium transporters.** Higher plants exhibit varied responses to ammonium ($NH_4^+$) and *S. lycopersicum* has been shown to be sensitive to $NH_4^+$ induced toxicity (reviewed in [31]). Fruit and leaf tissue produce ammonium though photorespiratory processes as well as incorporation of nitrogen from nitrate. Transcription as well as posttranslational regulation of the ammonium transporters is known to be upregulated under nitrogen deficiency [61]. Transporters implicated in ammonium translocation are categorized by the translocation fate of their primary substrate and can be found in almost every organism. The ammonium transporter/methylamine permease/rhesus (AMT1) family genes are involved in $NH_4^+$ uptake and $NH_4^+$ output and are thought to have evolved in the eukaryotes. Ammonium transporter/methylamine permease/rhesus (AMT2) genes of prokaryotic origin are involved in $NH_4^+$ uptake with $NH_3$ plus a proton as the output. Three members of the AMT1 family and AMT2.1 were detected in this study. *AMT1.1*, Solyc09g090730.2.1, had the highest expression values overall in the ORG and CONV leaf tissues (S11 Fig).

**Ammonium metabolism.** Conversion of toxic ammonium, whether generated through photorespiration or endogenous uptake, relies on glutamate synthase (GS) and decarboxylating glycine dehydrogenase (GDH). It has been shown in oilseed rape that a concurrent increase in $NH_4^+$ flux increased GS activity [62]. The cytosolic isoform of *GS* activity was moderately upregulated, log2FC 0.740, in the ORG leaf treatment corresponding to the results expressed in the Schjoerring, Husted (62) study (Fig 6D). Surprisingly, the plastidial *GS* isoform in the leaf treatment indicated the opposite trend, log2FC -0.587, and was barely detected in the fruit tissue (Fig 6A). The *NADP-GDH isoform 2* (Solyc01g068210.3.1) was the only isoform detected at appreciable levels other than the *GDH-B* isoform (S12 Fig).

**Amino acid transporters.** Nitrogen assimilation by plants includes uptake of amino acids (AA) and encode transporters specific to different groups of amino acids [63]. In this study, four major groups of AA transporter homologs specific to, the proline, amino acid permease, vacuolar amino acids, and the aromatic and neutral amino acids were detected. Two proline transporter transcripts, *ProT1* (Solyc03g096390.3.1) and *ProT3* (Solyc05g052820.3.1), were identified with expression of the former being higher than the latter which expressed at barely detectable levels. Neither of the proline transporters were differentially expressed between the ORG or CONV treatments. (S13A Fig). Three transcripts in the dataset were annotated as members of the amino acid permease group. *AAP2* (Solyc04g077050.3.1) and *AAP3* (Solyc11g005070.2.1) expression levels were barely detectable in all treatment and tissue samples. *AAPput* (Solyc07g066010.3.1), a putative homolog in the family, was moderately upregulated, log2FC 0.642, in the ORG leaf tissue and the most highly expressed among the three transcripts (S13A Fig). Four members of the vacuolar transporter group were detected in this study namely, two isoforms of *AAT1*, labelled as *vacAAT1a* (Solyc10g084830.2.1) and *vacAAT1b* (Solyc11g008440.2.1), *vacCAT2* (Solyc10g081460.2.1) and *vacCAT4* (Solyc02g037510.3.1). Two additional cationic transporters, *CAT5* (Solyc02g081850.3.1) was slightly upregulated in the ORG leaf tissue, and *CAT7* (Solyc11g006710.2.1) was present but at barely detectable levels. *vacCAT2* was the most highly expressed of the vacuolar transporter group with no discernable differences between the two treatments either in the fruit or leaf tissues. *vacAAT1b* exhibited the highest differential expression between the two tissues (S13A Fig). *ANT1lq* (Solyc03g117350.1.1) and the mitochondrial targeted *ANT1er* (Solyc06g069410.3.1) transcripts identified in this study belong to the aromatic and neutral amino acid transporter group. The expression values of *ANT1er* were the highest among the group with no discernable differences between the ORG and CONV treatments. *ANT1lq* expression values were very low and did not vary appreciably between tissue or treatment types (S13B Fig). Two ureide transporters,

# Nitrite related genes

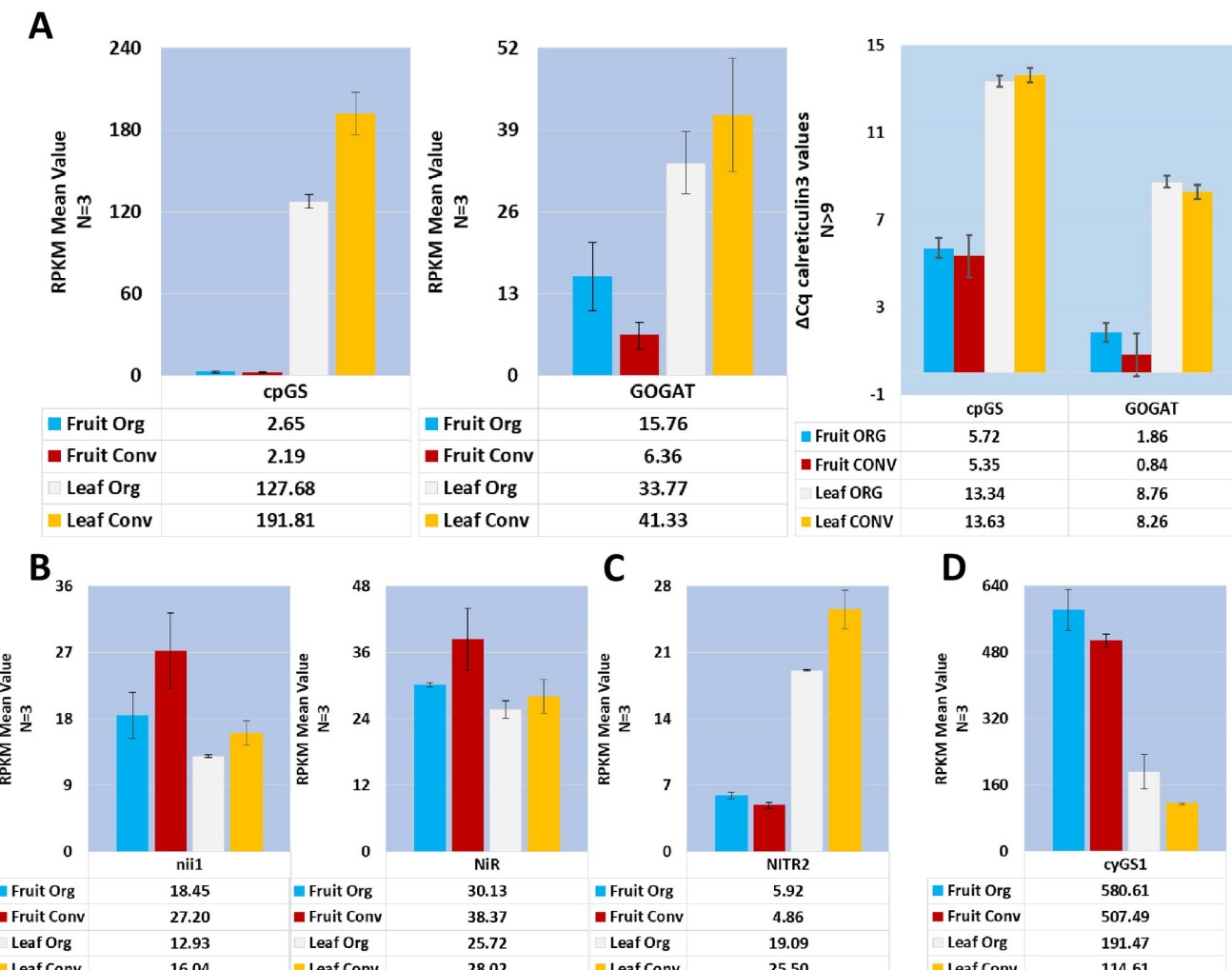

| Abbreviation | Sol Accession | Annotation | Lab Annotation | Org/CONV Fruit log2FC | ORG/CONV Leaf log2FC |
|---|---|---|---|---|---|
| cyGS1 | Solyc11g011380.2.1 | glutamine synthetase | contig 868 | 0.194190526 | 0.740419295 |
| cpGS | Solyc01g080280.3.1 | chloroplast glutamine synthetase | contig 48493 | 0.272247648 | -0.587203076 |
| cyGS2 | Solyc04g014510.3.1 | glutamine synthase | contig 24953 | -1.248084358 | -0.225459035 |
| GOGAT | Solyc03g083440.3.1 | Glutamate synthase, putative | contig 4595 | 1.308506676 | -0.291303711 |
| nii1 | Solyc01g108630.3.1 | nii1 nitrite reductase | contig 5372 | -0.5600249 | -0.310797649 |
| NiR | Solyc10g050890.2.1 | nitrite reductase 2 | contig 5373 | -0.348750039 | -0.123719563 |
| NITR2 | Solyc03g113280.3.1 | Integral membrane HPP family protein | contig 14570 | 0.283398599 | -0.417424275 |

**Fig 6. Expression profile of mean RPKM values of Nitrite Assimilation components in Fruit and Leaf tissue. A.** Chloroplast isoform of glutamine synthetase (cpGS) and glutamate synthase (GOGAT) RPKM and ΔΔCq values. **B.** RPKM values for two isoforms of Nitrate Reductase (nii1 and NiR) **C.** Identified homolog of plastidial nitrite transporter (NITR2). **D.** RPKM values for cytosolic isoform 1 of glutamine synthetase. Table of annotations and accession numbers from Sol Genomics ITAG3.2. Error bars denote the standard error of sequenced expression values, or in the case of cpGS and GOGAT RT-qPCR ΔΔCq values, between 3 biological replicates and at least 3 technical replicates.

*UPS1a* (Solyc01g010290.3.1) and *UPS1b* (Solyc04g005410.2.1), were detected with *UPS1a* expressed with the highest levels in the leaf tissue and at relatively even amounts between the treatment types in both tissues (S13B Fig).

## Nitrogen metabolism

**Nitrate reductase.** Irrespective of the type of transporter used, nitrate is reduced to nitrite by cytosolic nitrate reductase (NR). The expression values of *NR* (Solyc11g013810.2.1) were observed to be moderately down regulated, log2FC -0.633, in the sequenced ORG leaf treatment and slightly down regulated, $2^{-\Delta\Delta Ct}$ -1.3-fold change, in the RT-qPCR data. Overall fruit *NR* levels were a magnitude of order lower in comparison to the leaf tissues, with the *NR* levels in ORG fruit treatment being slightly upregulated, log2FC 0.315, in the sequencing data and moderately upregulated, $2^{-\Delta\Delta Ct}$ 0.707, in the RT-qPCR data (Fig 5A).

**Nitrite reductase.** Nitrite is transported into the plastids via the NITR family of transporters where it is converted to glutamate by ferredoxin dependent nitrite reductase. Two isoforms, *Nii1* and *NiR*, Solyc01g108630.3.1 and Solyc10g050890.2.1 respectively, of the plastid-targeted nitrite reductases were identified in the transcriptomic data. *Nii1* was moderately down regulated in the ORG fruit tissue, log2FC -0.550, and slightly down regulated, log2FC -0.310, in the ORG leaf tissue, whereas *NiR* was slightly down regulated, log2FC -0.349, in the ORG fruit tissue (Fig 6B).

**Glutamate/Glutamine–Glutamine synthetase.** Glutamate produced from nitrate reductase in the plastids is transported across the plasma membrane by the AMT1 family of transporters and converted to glutamine, in both cellular compartments, by glutamine synthetase (GS). Both cytosolic forms of GS, Solyc11g011380.2.1 (*cyGS1*) and Solyc04g014510.3.1 (*cyGS2*) and the plastidial Solyc01g080280.3.1 (*cpGS*), were identified in the transcriptome data. *cyGS1* exhibited the highest expression level of all isoforms in the fruit tissue and leaf tissue with *cpGS* expressed comparable levels to *cyGS1* in the leaf tissue. *cyGS1* was moderately upregulated, log2FC 0.740, in the ORG leaf tissue and *cpGS* was moderately downregulated, log2FC -0.587, in the ORG leaf tissue and barely detectable in the fruit tissues (Fig 6A & 6D).

Increased transporter expression for nitrogen uptake is indicative of higher nitrate consumption in the leaf tissue relative to the fruit tissue as the fruit tissue transcriptional activity was scarcely detected. The NRT1 family of genes is known to be constitutively expressed in plants [58, 61]. Additionally, it can be inferred that the fruit tissue nitrate requirements were low from the lack of transcriptional activity exhibited by the NRT1 family of transporters. As nitrate is not utilized by the plant directly, but incorporated as ammonium, transcriptional activity, associated with the enzymes and transporters from nitrate to nitrite to ammonium, was higher in the leaf CONV treatments with the exception of the nitrite reductase isoforms expressed at marginally higher levels in the fruit CONV tissue and treatment and a highly expressed cytosolic glutamine synthetase isoform in the fruit ORG tissue (Figs 5 & 6). Leaf ammonium and glutamine levels have been shown to have a negative correlation to nitrate reductase transcription activity [56]. Transcript levels of nitrate reductase in the leaf ORG and both fruit tissue treatments implicate higher levels of ammonium, glutamate and glutamine in these tissues. The transcript activity differential between glutamine synthetase and glutamate dehydrogenase suggest the metabolic flux runs in favor of the production of glutamate.

Additionally, the increase of glutamate would add to the increase of umami flavor, as well as the significant increase in soluble solids that contribute to consumer taste preference of tomatoes [64].

## Photosynthetic/Respiratory processes

The Gene Ontology enrichment of the transcriptome data revealed that the photosynthetic components and processes, which include photorespiratory and respiratory components and processes, were enriched in the ORG fruit tissues while the translational machinery and non-specific organelle components were enriched in the CONV fruit tissue. This may be somewhat surprising since mature fruit tissues are not expected to perform any photosynthesis. However, GO terms implicated in photosynthesis are included in the photorespiratory and respiration processes. Analysis of the annotations of the enriched GO terms revealed that enzymes involved in the TCA cycle were enriched while the expression of transcripts coding for chlorophyll binding protein was reduced in the ORG treatment.

Genes related to the plant's photosynthetic processes; the ATP-dependent zinc metalloprotease FtsH in chloroplast development [65], photosystem I subunit B (psaB) and photosystem II subunit A (psbA) related to the electron transport chain [66], ribulose-1,5-bisphosphate carboxylase/oxygenase (rubisco) large subunit (rbcL) for carbon/oxygen fixation [67], peroxisomal (S)-2-hydroxy-acid oxidase (glycolate oxidase, GLO) involved in remediation of the fixed oxygen via photorespiration [68], and the superoxide dismutase, L-ascorbate peroxidase (APX) and catalase genes [69] involved in the scavenging of reactive oxygen species are useful in understanding the photosynthetic status of the plant. Expression of the genes involved in the capture and conversion of energy, psbA, psaB and rbcL, and chloroplast duplication, FtSHcp, were non-differentially expressed. The exception was psaB, which was slightly down regulated, log2FC -0.398, in the ORG leaf tissue and slightly up regulated in the ORG fruit tissue (S14 Fig). Conversely, APX.1 and SOD Cu-Zn.1, significantly and moderately upregulated respectively, the genes related to photosynthetic stress remediation of ROS scavenging demonstrated an increased expression in ORG leaf (Fig 3), implying the ORG treated plants were under photosynthetic oxidative stress. The antioxidative metabolites produced during photosynthetic stress are known to be photosynthetic feedback regulators and known to inhibit the expression of photosynthetic genes. In addition to the increase in these genes, the decrease in photosystem I and photosystem II reaction center transcripts, psaB and psbA (S14 Fig) is an additional indicator that the ORG treated plants did not hold as much capacity for photosynthetic activity as the CONV treated plants did. These observations indicate that the plants under the ORG treatment were operating under photosynthetic stress conditions where a reduction in chlorophyll content and increase in the reactive oxygen species (ROS) scavenging genes could be used as mechanisms to reduce production of ROS. Under such conditions, plants attempt to decrease photosystem II activity by activating the Mehler peroxidase reaction (reviewed in [70]), which results in an increase in superoxide radicals. Increased oxygen radicals stimulate SOD production [71, 72], and APX and catalase expression activity during tomato fruit ripening [73]. Increased level of APX expression was also observed in *Lactuca sativa* L. cv Romaine leaves when Mehler reaction was initiated by potassium cyanide (KCN)-mediated inhibition of the electron transport chain [74]. In this study, differential increase of the *APX* and *cpSOD Cu-Zn* isoforms in ORG leaf and upregulation of *Psy1* in ORG leaf discussed previously imply that ORG treatment produces a photoinhibitory environment for the plant.

The TCA cycle is known to operate in a closed mode, where the cycle functions normally with all the enzymes operating to produce chemical energy and amino acid precursors, or in

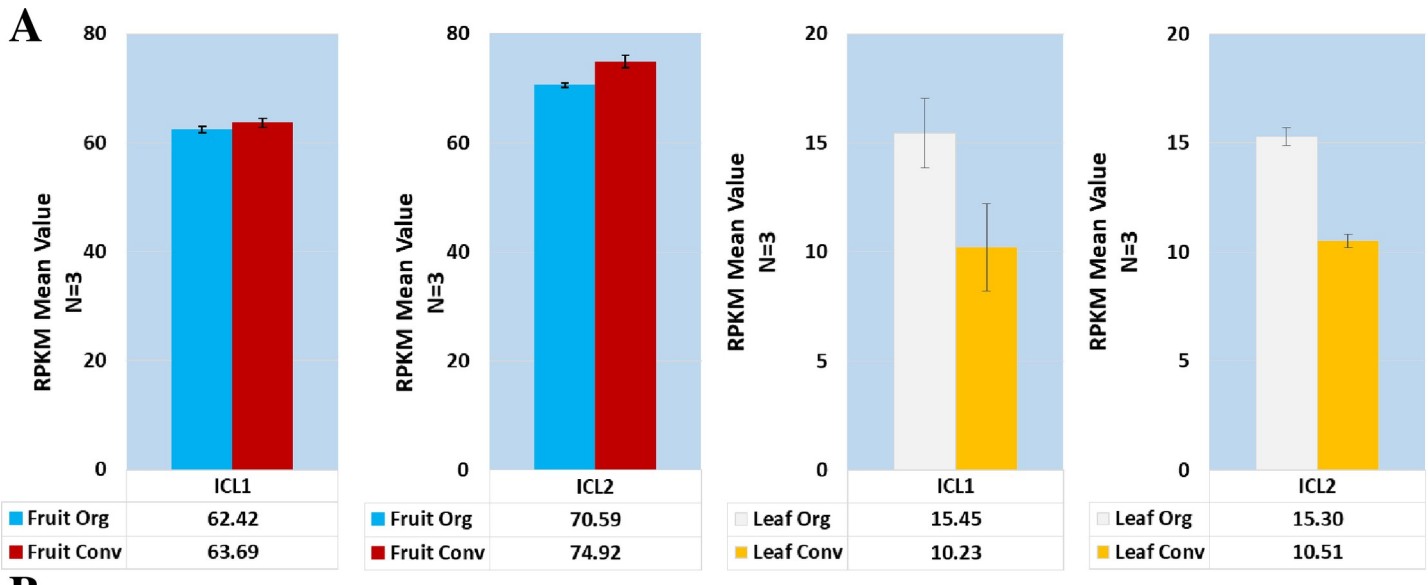

**Fig 7. Differential expression of isocitrate lyase homologs. A.** Mean expression values for the identified isocitrate lyase isoforms. **B.** Table of annotations and accession numbers from Sol Genomics ITAG3.2. Error bars denote the standard error of sequenced expression values between 3 biological replicates.

an open mode leading to the accumulation of different organic acids dependent on where the flux break occurs. These breaks can occur due to changes in redox status or under photorespiratory conditions [75]. When under photosynthetic stress, a notable flux break results in the glyoxylate shunt in the TCA cycle where metabolite flux is shunted from isocitrate through glyoxylate to malate. An indication this could occur in the ORG leaf tissue is from the expression of isocitrate lyase (ICL). Two isoforms (Solyc07g052480.3.1, *ICL1*,and Solyc05g056270.3.1, *ICL2*) were present in the data. While the fruit tissue did not display differential expression, the leaf ORG tissue was moderately upregulated in both *ICL1* and *ICL2* with log2FC values of 0.595 and 0.543 respectively (Fig 7). The alterations in gene expression may be responsible for allocation of metabolite to the production of organic acids in the ORG tissue while the TCA cycle metabolites in the CONV tissue route through α-ketoglutarate for amino acid biosynthesis.

In the context of ripening and respiration, the malic enzymes (ME) and family of malate dehydrogenases (MDH) are responsible for the decarboxylation and/or oxidation of accumulated malate pools to pyruvate and OAA respectively (S15B & S15C Fig). In this transcriptome data, seven *MDH* isoforms, targeted to different cellular locations; cytosol–*MDH1* (Solyc09g090140.3.1) and *MDH-like 2.1* (Solyc09g091070.2.1) and *MDH-like 2.2* (Solyc03g115990.2.1), glyoxysome– *glyMDH3.1* (Solyc02g063490.3.1) and *glyMDH3.2* (Solyc01g106470.3.1), and chloroplast– *cpNADPMDH1* (Solyc11g007990.2.1) and *cpNADPMDH 2* (Solyc03g071590.3.1), were identified.

The *MDH1* transcripts were present at the highest levels in the leaf tissue and the *glyMDH3.1* iso-form in the fruit tissue compared to the other *MDH* isoforms. The *glyMDH3.1* transcript was highly abundant in fruit tissues although not differentially expressed but was significantly higher, log2FC 1.00, in the ORG leaf tissue (Fig 8 and S15C Fig).

## Conclusions

In this study two hypotheses were evaluated. The first hypothesis that organic fertilizer, which results in a slower rate of biological release of available nitrogen to plant roots, results in greater allocation of photosynthetically derived resources to the synthesis of secondary metab-olites, such as phenolics and other antioxidants, than to plant growth, was supported by the higher accumulation of lycopene, ascorbate and TEAC. The second hypothesis stated that the genes involved in changes in the accumulation of phytonutrients under organic fertilizer regime will exhibit differential expression, and that the growth under different fertilizer treat-ments will elicit a differential response from the tomato genome, was partially supported. Based on the metabolic pathway modeling presented in this study, we propose that the organic fertilizer treatment results in activation of photoinhibitory processes through differential acti-vation of nitrogen transport and assimilation genes resulting in higher accumulation of the phytonutrients quantified in the fruit tissues. These data also provide information regarding which genes are impacted the most due to the differences in source of fertilizer. This knowl-edge can be used to identify alleles that allow for efficient utilization of organic inputs for breeding crops that are acclimatized to organic inputs.

## Materials and methods

**Plant material.**   Tomato seeds (*Solanum lycopersicum* L.) 'Oregon Spring' sourced from Johnny's Selected Seeds, Winslow, ME, were sown in LC1 Professional Growing Mix (Sun Grow Horticulture, Bellevue, WA). Glasshouse temperatures were maintained at 21.1/18.3ºC (day/night) with a 14 h day, supplemented with high-pressure sodium lamps and 10h night photoperiod. Three-week emergent seedlings were fertilized with a BioLink All Purpose Fertil-izer 5-5-5 (Westbridge Agricultural Products, Vista, CA) solution at a concentration of 4 mL/L tap water representing the organic nutrient treatment (ORG). The seedlings grown under con-ventional fertilizer (CONV) received a Peters 20-10-20 solution (1.02 g/L tap water). Both treatments received one liter per week of their respective nutrient treatment.

Six-weeks-old plants of similar size and vigor from each treatment were transferred to indi-vidual 24-liter pots. Potting media for the ORG nutrient treatment consisted of a mix of LC1 Professional Growing Mix (Sun Grow Horticulture, Bellevue, WA), Whitney Farms Compost (Scotts, Marysville, OH), and sifted soil in a ratio by volume of 15:5:1. The CONV plants were potted in 100 percent LC1 Professional Growing Mix. Upon transplantation, individual plants were fertilized with 150 mL of applicable nutrient solution once per week. Twelve plants in each nutrient treatment were selected at week seven and arranged in a randomized complete block design consisting of six blocks. Plants were provided 1 liter of water every other day and the dosage of weekly nutrient treatments was increased to 500 mL beginning at week eight. Lateral shoots below the first flower cluster were removed as they appeared. Starting in week 11, the CONV nutrient solution concentration was increased from 1.02 g/L of Peters 20-10-20 to 1.25 g/L. Beginning in week 12, nutrient treatments were augmented with micronutrients. The ORG treatment was amended with BioLink Micronutrient Fertilizer (Westbridge Agricul-tural Products, Vista, CA) at 3.9 mL/L tap water and the CONV treatment was augmented with calcium phosphate monobasic monohydrate at 166 mg/L tap water. Concentrations of macronutrients were equivalent in both treatments, with total nitrogen, total phosphorus and

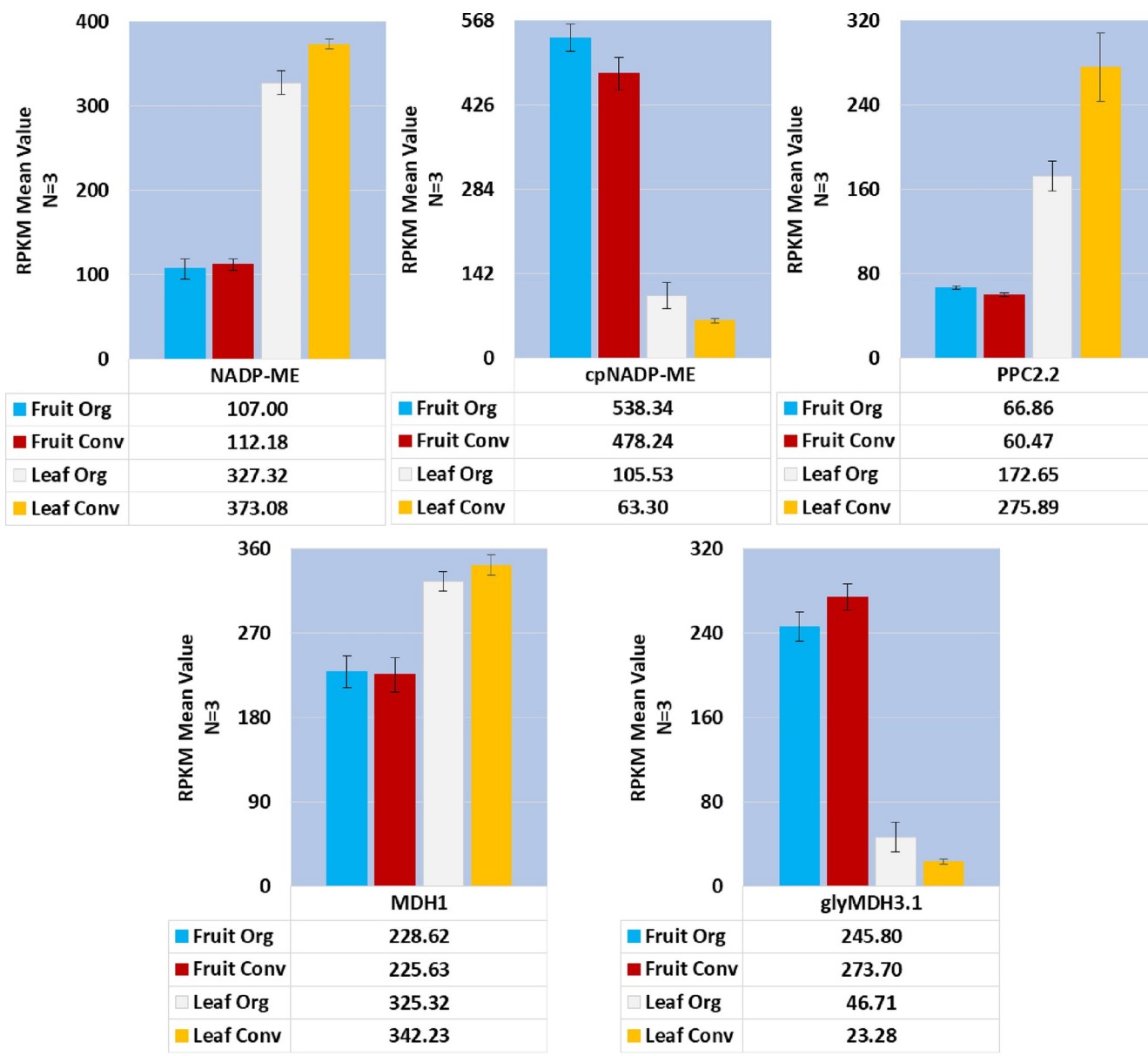

| Abbreviation | Sol Accession | Annotation | Lab Annotation | Org/CONV Fruit log2FC | ORG/CONV Leaf log2FC |
|---|---|---|---|---|---|
| NADP-ME | Solyc05g050120.3.1 | cytosolic NADP-malic enzyme | Contig 546 | -0.068091488 | -0.18879947 |
| cpNADP-ME | Solyc12g044600.3.1 | NADP-malic enzyme | Contig 379 | 0.170780839 | 0.737418414 |
| PPC2.2 | Solyc07g055060.3.1 | Phosphoenolpyruvate carboxylase | contig 770 | 0.170780839 | -0.67623855 |
| MDH1 | Solyc09g090140.3.1 | Malate dehydrogenase* sol annot to ferridoxin | contig 1114 | 0.01902863 | -0.073138786 |
| glyMDH3.1 | Solyc02g063490.3.1 | glyoxisomal malate dehydrogenase | contig 3340 | -0.15508352 | 1.004610062 |

**Fig 8. Mean RPKM expression values of enzymes involved in photosynthetic and/or tricarboxylic acid cycle respiratory pathway genes.** Table of annotations and accession numbers from Sol Genomics ITAG3.2. Error bars denote the standard error of sequenced expression values between 3 biological replicates.

total potassium concentrations of 260 ppm, 99 ppm and 235/220 ppm (CONV/ORG respectively) based on laboratory analyses (Analytical Science Laboratory, University of Idaho, Moscow, ID). Nitrogen forms varied greatly between CONV and ORG treatments with nitrate constituting the dominant form in CONV, and organic forms (i.e. amino acids and proteins) predominating the ORG nutrient solution. In week 13, the air temperature in the glasshouse was increased to 23.3/20ºC (day/night) and the nutrient dosage was increased to 750 mL per plant every other day. In week 14 the nutrient dosage was increased to 1 liter every other day.

Fruit were harvested at ripe red stage beginning in week 18 and continuing until a minimum of eight ripe fruit were harvested from each plant during week 23. Immediately following harvest, all tomato fruit were weighed and their stages of maturity noted based on previously established standards [76]. A 1 cm thick slice was cut from the equatorial region of each fruit and the pericarp tissue external to each locule was sampled. The tissue was minced with a knife, one portion of which was used to measure percent soluble solids (ºBrix) using a digital refractometer (Atago PR-101, Bellevue, WA), while the remainder was flash-frozen with liquid nitrogen, placed in 50-mL plastic tubes, and stored in a -80ºC freezer for biochemical and transcriptome analysis. The samples were ground by mortar and pestle in liquid nitrogen until finely powdered and used for phytochemical analyses. Analyses of total carbon and total nitrogen (via combustion) and calcium, potassium, magnesium, sodium, phosphorus and sulfur (ICP with nitric digestion) were conducted on dried leaf tissues (Analytical Science Laboratory, University of Idaho, Moscow, ID).

Total phenolic compounds were measured spectrophotometrically using the method outlined by Singleton et al. (1999) [47] with modifications. One milliliter of an 80% methanol solution was added to frozen, 350 mg samples of powdered fruit tissue or 150 mg of powdered leaf tissue in 2 mL microcentrifuge tubes, with all samples run in triplicate. Tubes were vortexed for 30 sec until thoroughly dispersed and stored at -20˚C for 24 hr. Samples were centrifuged at 8,000 g and 4˚C for 20 min. The supernatant from each tube was then poured into individual 15 mL polyethylene sample tubes, capped and stored at -20˚C. The above extraction was conducted two more times on the same tissue. The supernatant was adjusted to 3 mL with 80% methanol. Then, 200 µL of extract, 1000 µL of 10% Folin-Ciocalteu (F-C) phenol reagent, 2N and 800 µL of 75 g L-1 $Na_2CO_3$ were pipetted into 2 mL microcentrifuge tubes, vortexed and stored at 20˚C for 2 hr. A second solution with each extract was made substituting 800 µL of Nanopure water for the $Na_2CO_3$ and stored at 20˚C for 2 hr. An ultraviolet-visible spectrophotometer (Agilent Technologies, Model HP8453, Avondale, PA) interfaced to a computer with UV-Visible ChemStation software (v. B.01.01) with tungsten bulb was blanked on 1.0 mL of 10% F-C reagent in a 1.5 mL polystyrene cuvette. One milliliter of the sample solutions was pipetted into separate cuvettes and absorbance was measured at 760 nm. Cuvettes were run in duplicate. The average difference in absorbance was compared to a standard curved for gallic acid, and the concentrations were reported as gallic acid equivalents (GAE).

Lycopene was measured spectrophotometrically using the method outlined by Nagata and Yamashita (1992) [77] with modifications. The following work was conducted under low light. Powdered, frozen fruit samples of 100 mg were placed in 15 mL conical polyethylene sample tubes wrapped with aluminum foil with all samples run in triplicate. To each tube, 8.33 mL of a 2:3 solution of HPLC-grade acetone and HPLC-grade hexanes at 4˚C was added. The solution was amended with $9.08 \times 10^{-4}$ mol L-1 (200 mg L-1) butylated hydroxytoulene (BHT) to

prevent pigment oxidation. Tubes were capped and vortexed for 1 min, then stored at -20˚C for 24 hr. After storage the tubes were again vortexed for 1 min and returned to -20˚C. This was continued for a total of 96 hr. of extraction. Tubes were vortexed a final time and let stand for 5 min for a complete phase separation to occur. The spectrophotometer was blanked and zeroed using a tungsten lamp and 1 mL 100% HPLC-grade hexanes in a 1.4 mL glass cuvette at 505 nm. Aliquots of 1 mL of the hexane phase were pipette into 1.4 mL glass cuvettes and absorbance was measured at 505 nm. Lycopene concentration was calculated according to the equation of Davies (1976): Lycopene (mM) = Absorbance505/3400 mM -1

For transcriptome analysis, samples were processed via pulverization under liquid nitrogen in the SPEX SamplePrep® FreezerMill 6870 (Metuchen, NJ USA). Three to four randomly selected representative leaves were harvested at week 23 as sub-samples from the mid to upper canopy of all plants, flash-frozen with liquid nitrogen and stored in a -80ºC freezer and pulverized in the SPEX SamplePrep® FreezerMill 6870 (Metuchen, NJ USA) prior to extraction of RNA.

**RNA extraction.** RNA was isolated from CONV and ORG tomato fruit and leaf tissue at the red ripe stage with the QIAGEN RNeasy Plant Mini Kit (Valencia, CA). Extractions were performed by combining three to four fully expanded leaves from the mid to upper canopy (leaf samples) or three red ripe fruit as they became ripe (fruit samples) from two plants constituting one biological replicate. RNA was extracted from each biological replicate; three CONV fruit, three ORG fruit, three CONV leaf and three ORG leaf were used as input for sequencing and RT-qPCR procedures. RNA concentration was determined for each extraction of each unique treatment-sample combination using a Nanodrop ND-8000 (ThermoFisher, MA, USA) and sample integrity was validated using denaturing agarose gel electrophoresis. The samples were treated with DNAse using the Ambion TURBO DNA-free kit (ThermoFisher, MA, USA) to remove any potential DNA contamination. RNA integrity and concentration were verified for all samples following DNAse treatment.

**Illumina sequencing.** RNA extractions were used to generate twelve individually barcoded sequencing libraries with Illumina's TruSeq RNA Sample Preparation v2 kit (San Diego, CA, USA) with minor modifications. Modifications to the published protocol include a decrease in the mRNA fragmentation incubation time from 8 minutes to 30 seconds to create the final library proper molecule size range. Additionally, Aline Biosciences' (Woburn, MA, USA) DNA SizeSelector-I bead- based size selection system was utilized to target final library molecules for a mean size of 450 base pairs. The quantity and quality of all the libraries were then ascertained using Life Technologies (Carlsbad, CA, USA) Qubit Fluorometer and an Agilent (Santa Clara, CA, USA) 2100 Bioanalyzer (Dr. Jeff Landgraf, Michigan State University, personal communication). The Illumina Hi Seq 2000 sequencing platform (San Diego, CA, USA) was used to sequence the cDNA libraries as 2x100 PE reads across three lanes of a flow-cell at Michigan State University's Research Technology Support Facility. Read files were submitted to the National Center for Biotechnology Information (NCBI) Short Read Archive (SRA) database under SRR4102059 (leaf conventional treatment), SRR4102061 (leaf organic treatment), SRR4102063 (fruit conventional treatment), and SRR4102065 (fruit organic treatment).

**Data processing, assembly, identification of differentially expressed genes and visualization of genome-wide expression.** Sequence read information from Illumina HiSeq 2000 2x100 PE fastq files were used as input for the CLC Bio Genomic Workbench (ver 6.0.1) (Aarhus, Denmark). All read datasets were processed with the CLC Create Sequencing QC Report tool to assess read quality. The CLC Trim Sequence process was used to trim the first 12 bases due to GC ratio variability and for a Phred score of 30. All read datasets were trimmed of ambiguous bases. Illumina reads were then processed through the CLC Merge Overlapping

Pairs tool and all reads were *de novo* assembled to produce contiguous sequences (contigs). Mapped reads were used to update the contigs and contigs with no mapped reads were eliminated from the dataset. Non-trimmed reads used for assembly were mapped back to the assembled contigs. Consensus contig sequences were extracted as a multi-fasta file. The individual CONV and ORG read datasets, original non-trimmed reads, were mapped back to the assembled contigs to generate individual treatment sample reads per contig and then normalized with the Reads Per Kilobase per Million reads (RPKM) method [78] and the log2 Fold Change values were calculated from RPKM value ratios.

The ORG/CONV RPKM value ratios, calculated from loci mapped on the *S. lycopersicum* genome build 2.40 chromosomal maps provided by the Sol Genomics Network [79], obtained from the RNAseq data analysis were $\log_{10}$ transformed and the comparison between ORG and CONV treatments for each discrete area on the chromosome was visualized via Manhattan plots. At any given position on the chromosome, red bars indicate a higher $\log_{10}$ ratio in the CONV treatment, and blue bars indicate higher expression in the ORG treatment. Trend lines were graphed to indicate a predominance of chromosomal activity either in the ORG treatment (positive y-intercept value) or in the CONV treatment (negative y-intercept value).

**Functional annotation.** Assembled contiguous sequences (contigs) were annotated by alignment with blastx through Blast2GO [22] (BioBam Bioinformatics S.L., Valencia, Spain) as well as local stand-alone blastx alignments against the NCBI nr database (ver. 2.2.27+) [80] and blastn alignments against the ITAG3.2_CDS.fasta file downloaded from the Sol Genomics Network FTP site (ftp://ftp.solgenomics.net/genomes/Solanum_lycopersicum/annotation/ITAG3.2_release/). Gene ontology (GO) annotation, enzyme code annotation and the EMBL-EBI InterProScan annotation of predicted protein signatures were all annotated through Blast2GO [21, 22]. The BLAST annotated RNA-Seq datasets from the conventional and organic treatments were analyzed for GO enrichment with Blast2GO [21]. Unless otherwise specified, expression analysis was restricted to the contig consensus sequence annotation and does not represent specific alleles, gene family members of highly similar sequence or subunit specificity [81].

Fisher's exact test was performed to identify any nonrandom associations between the Gene Ontology terms associated with the annotated transcripts to identify any over- or underrepresented GO terms involved in the phytonutrient pathways related to TEAC, total phenolics, and soluble solids. ORG/CONV RPKM ratios were $\log_{10}$ converted and separated as 2-fold greater, $\log_{10}$ value less than -0.3010 –higher in the CONV treatment or $\log_{10}$ greater than 0.3010 –higher in the ORG treatment, between the treatments and non-differential, $\log_{10}$ values between -0.3010 and 0.3010 (S8 and S9 Figs). Contig read counts were included in the Blast2GO analysis and queried with Fisher's exact test for differential expression within a General Linearized Model (GLM), and the upper quartile between ORG and CONV leaf and fruit treatments.

**Real-time quantitative PCR (RT-qPCR).** RNA extractions used to generate the twelve individually barcoded sequencing libraries, one from each biological replicate, were utilized for first-strand cDNA synthesis for subsequent quantitative polymerase chain reactions using the Invitrogen SuperScript VILO kit in 20μl reactions. Product integrity was checked using agarose gel electrophoresis. Concentration for each cDNA preparation was evaluated using a Qubit fluorimeter (Life Technologies–Carlsbad, CA, USA). Real-time quantitative PCR (RT-qPCR) was performed in triplicate from each unique cDNA preparation derived from three independent experiments. The abundance of 19 differentially expressed genes selected from *in silico* RPKM analysis of RNAseq data was quantified. Each reaction was loaded with 25 ng of template first-strand cDNA, and tested on the Stratagene MX3005P light cycler (Agilent Technologies–Santa Clara, CA) using the iTaq Universal SYBR Green Supermix reagent (with ROX

passive dye) (Bio-Rad Laboratories–Hercules, CA). Reaction conditions and thermal profile can be found in the recommended protocols provided by the manufacturer. Instrument fluorescence was used as an input for the LinRegPCR tool [82, 83] and analyzed using the plate-wide-mean mode for extraction of reaction Cq and efficiency values. Fold-change values of transcript abundance were calculated relative to the geometric mean of internal control of cal-reticulin3-like transcripts and represented using the $2^{-\Delta\Delta Ct}$ fold change calculations and Pfaffl-method correction [84, 85].

## Supporting information

**S1 Table. Above and below ground vegetative biomass on fresh (FW) and dry weight (DW) bases, and percent root biomass fraction for conventional (CONV) and organic (ORG) fertilizer treatments.**
(DOCX)

**S2 Table. Leaf dry matter and mineral concentrations on a dry weight (DW) basis for conventional (CONV) and organic (ORG) fertilizer treatments.**
(DOCX)

**S3 Table. Sequenced and assembled reads.**
(GIF)

**S4 Table. Sol Genomics Network ITAG ver. 3.2 CDS annotations to gene lab abbreviations from S5 Fig.**
(GIF)

**S5 Table. Sol Genomics Network ITAG ver. 3.2 CDS annotations to gene lab abbreviations from S6 Fig.**
(GIF)

**S1 Fig. Mean red ripe fruit mass under conventional (CONV) and organic (ORG) fertilizer treatments.**
(TIF)

**S2 Fig. RT-qPCR reference gene validation across tissue and treatment types.**
(TIF)

**S3 Fig. ORG and CONV log10 RPKM ratio Manhattan plot representations of expression values across S. lycopersicum chromosomes 1 to 12.**
(TIF)

**S4 Fig. Mean expression values of Lycopene Biosynthesis Pathway Enzymes.**
(TIF)

**S5 Fig. Mean expression values of the Smirnoff-Wheeler pathway enzymatic transcript activity.**
(TIF)

**S6 Fig. Mean expression values of the Foyer-Halliwell-Asada Pathway enzymatic transcript activity.**
(TIF)

**S7 Fig. Differential expression breakout of contigs assigned to the enriched GO terms.**
(TIF)

**S8 Fig.** Fruit GO terms concluded to be differentially expressed by the general linear model (A) and for the upper quartile (B).
(TIF)

**S9 Fig.** Leaf GO terms concluded to be differentially expressed by the general linear model (A) and for the upper quartile (B).
(TIF)

**S10 Fig. Mean expression values of the Phosphate uptake indicator genes.**
(TIF)

**S11 Fig. Mean expression values of Ammonium Assimilation transporters in Fruit and Leaf tissue.**
(TIF)

**S12 Fig. Mean expression values of the Glutamate Dehydrogenase genes.**
(TIF)

**S13 Fig. Detected and identified amino acid transporters and mean expression values.**
(TIF)

**S14 Fig. Mean expression values of Photosynthesis Indicator genes.**
(TIF)

**S15 Fig. Photorespiratory and/or Respiratory pathway genes.**
(TIF)

## Acknowledgments

This research was funded in part by CSANR BIOAg grant to PA and AD, and by the USDA National Institute of Food and Agriculture, Hatch project WNP00011 to AD. SLH and BRK acknowledge the support of NIH/NIGMS through institutional training grant award T32-GM008336. The contents of the publication are solely the responsibility of the authors and do not necessarily represent the official views of the NIGMS or NIH.

## Author Contributions

**Conceptualization:** Luke Gustafson, Preston Andrews, Amit Dhingra.

**Data curation:** Richard M. Sharpe, Luke Gustafson, Seanna Hewitt, James Crabb, Christopher Hendrickson, Derick Jiwan, Amit Dhingra.

**Formal analysis:** Richard M. Sharpe, Luke Gustafson, Seanna Hewitt, Benjamin Kilian, James Crabb, Christopher Hendrickson, Derick Jiwan, Preston Andrews, Amit Dhingra.

**Funding acquisition:** Preston Andrews, Amit Dhingra.

**Methodology:** Richard M. Sharpe, Luke Gustafson, Derick Jiwan, Preston Andrews, Amit Dhingra.

**Project administration:** Preston Andrews, Amit Dhingra.

**Resources:** Preston Andrews, Amit Dhingra.

**Software:** Richard M. Sharpe, Amit Dhingra.

**Supervision:** Preston Andrews, Amit Dhingra.

**Validation:** Richard M. Sharpe, Luke Gustafson, Amit Dhingra.

**Visualization:** Richard M. Sharpe, Luke Gustafson, Amit Dhingra.

**Writing – original draft:** Richard M. Sharpe, Luke Gustafson, Seanna Hewitt, Benjamin Kilian, James Crabb, Christopher Hendrickson, Derick Jiwan, Preston Andrews, Amit Dhingra.

**Writing – review & editing:** Richard M. Sharpe, Luke Gustafson, Seanna Hewitt, Preston Andrews, Amit Dhingra.

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
