## [Decision Letter · Decision Letter 0]

27 Sep 2019

PONE-D-19-25332

Concomitant phytonutrient and transcriptome analysis of mature fruit and leaf tissues of tomato (Solanum lycopersicum L. cv. Oregon Spring) grown using organic and conventional fertilizer

PLOS ONE

Dear Professor Dhingra,

Thank you for submitting your manuscript to PLOS ONE. After careful consideration, we feel that it has merit but does not fully meet PLOS ONE’s publication criteria as it currently stands. Therefore, we invite you to submit a revised version of the manuscript that addresses the points raised during the review process.

We would appreciate receiving your revised manuscript by Nov 11 2019 11:59PM. To enhance the reproducibility of your results, we recommend that if applicable you deposit your laboratory protocols in protocols.io, where a protocol can be assigned its own identifier (DOI) such that it can be cited independently in the future. For instructions see: http://journals.plos.org/plosone/s/submission-guidelines#loc-laboratory-protocols

We look forward to receiving your revised manuscript.

Kind regards,

Yuan Huang

Academic Editor

PLOS ONE

Journal Requirements:

1. We note that you have included the phrase “data not shown” in your manuscript. Unfortunately, this does not meet our data sharing requirements. PLOS does not permit references to inaccessible data. We require that authors provide all relevant data within the paper, Supporting Information files, or in an acceptable, public repository. Please add a citation to support this phrase or upload the data that corresponds with these findings to a stable repository (such as Figshare or Dryad) and provide and URLs, DOIs, or accession numbers that may be used to access these data. Or, if the data are not a core part of the research being presented in your study, we ask that you remove the phrase that refers to these data.

Reviewers' comments:

Reviewer's Responses to Questions

**Comments to the Author**

1. Is the manuscript technically sound, and do the data support the conclusions?

Reviewer #1: Yes

2. Has the statistical analysis been performed appropriately and rigorously? 

Reviewer #1: Yes

3. Have the authors made all data underlying the findings in their manuscript fully available?

Reviewer #1: No

4. Is the manuscript presented in an intelligible fashion and written in standard English?

Reviewer #1: Yes

5. Review Comments to the Author

Reviewer #1: This manuscript presenting some valuable information and has sufficient level of novelty, this may be published after necessary revision.

Page 7: “In this study, less than a 9% difference in the transcriptional abundance of GPP1 between the fruit tissues was observed” Normally differential expression is mentioned based on Log2 fold change, you have calculated the difference of expression in percentages, apparently 9% seems an obvious change however if you calculate based on log2 fold change the difference might not be so obvious, please check correct accordingly.

Please represent the changes (up-regulation and down-regulation) in gene expressions/transcripts based on log2 fold change rather fold change or percentages (throughout the document).

Page 5: Please replace “fertility treatments” with “fertilizer treatment”

The manuscript seems overly long, please make it comparatively shorter.

Another observation is that throughout the manuscript reader has focused to highlight or show the up-regulated genes however the information or description of down-regulated genes has not got due attention.

Conclusion section is also long, this must not be more than 150-200 words.

Probably the raw data of transcriptome has not been uploaded to any publicly available platform such as NCBI or any other, please upload raw data and mention accession number in the manuscript.

This article may be cited in the discussion as this is closely related with the current study "19. Muhammad Azher Nawaz, Chen Chen, Fareeha Shireen, Zuhua Zheng, Hamza Sohail, Muhammad Afzal, Muhammad Amjad Ali, Bie Zhilong, Yuan Huang. 2018. Genome-wide expression profiling of leaves and root of watermelon in response to low nitrogen. BMC Genomics. 19: 456 " https://bmcgenomics.biomedcentral.com/track/pdf/10.1186/s12864-018-4856-x

6. PLOS authors have the option to publish the peer review history of their article (what does this mean?). If published, this will include your full peer review and any attached files.

Reviewer #1: Yes: Muhammad Azher Nawaz

---

## [Author Response · Author response to Decision Letter 0]

27 Oct 2019

Dear Editor, 

The authors are thankful to the reviewers for their critical and supportive comments. We have considered each comment and have made revisions to the manuscript along with responding to the comments. A point-by-point response to the reviewer’s comments is provided below. The suggested revisions have improved the manuscript and we hope that the manuscript will now be acceptable for publication in PLOS One. 

Editorial Comment: We note that you have included the phrase “data not shown” in your manuscript. Unfortunately, this does not meet our data sharing requirements. PLOS does not permit references to inaccessible data. We require that authors provide all relevant data within the paper, Supporting Information files, or in an acceptable, public repository. Please add a citation to support this phrase or upload the data that corresponds with these findings to a stable repository (such as Figshare or Dryad) and provide and URLs, DOIs, or accession numbers that may be used to access these data. Or, if the data are not a core part of the research being presented in your study, we ask that you remove the phrase that refers to these data.

Response: The phrase “data not shown” was an oversight in the original submission. All data are included in the main body or in the supplementary section of the revised submission. 

Comment #1: This manuscript presenting some valuable information and has sufficient level of novelty, this may be published after necessary revision.

Response: Authors appreciate the reviewer’s overall support. 

Comment #2: Page 7: “In this study, less than a 9% difference in the transcriptional abundance of GPP1 between the fruit tissues was observed” Normally differential expression is mentioned based on Log2 fold change, you have calculated the difference of expression in percentages, apparently 9% seems an obvious change however if you calculate based on log2 fold change the difference might not be so obvious, please check correct accordingly. Please represent the changes (up-regulation and down-regulation) in gene expressions/transcripts based on log2 fold change rather fold change or percentages (throughout the document).

Response: The authors agree with the reviewer that normally differential expression is mentioned based on log2 fold-change. The focus of this study, as stated in the hypothesis, was that the genes involved in pathways that corresponded to the phytonutrient differences (Lycopene and Ascorbate pathways), would be differentially expressed. However, we didn’t find support for the hypothesis, perhaps due to the fact that we analyzed only the mature leaf and fruit tissues. 

When we performed GO (gene ontology) enrichment, we observed the terms that were differentially enriched between ORG vs CONV. In the ontologies that showed significant enrichment according to the Fisher’s Exact Test, we identified the associated genes and their RPKM values. Thereafter, we analyzed the changes in RPKM values of the said genes in the context of the pathways they participate in. This methodology is detailed in the methods section. We refrained from stating that any of these genes were significantly differentially expressed unless the RPKM values were doubled from one treatment to the other, which then would be considered to be differentially expressed using the log2 fold-change parameter. 

We used this approach to understand the relationships between the pathway enzymes. Therefore, the data are represented as percentage based on RPKM values. However, if the editor considers we should represent the data in terms of significant changes only, we will be happy to make the necessary edits. 

Comment #3: Page 5: Please replace “fertility treatments” with “fertilizer treatment”

Response: The manuscript has been edited as advised. 

Comment #4: The manuscript seems overly long, please make it comparatively shorter.

Response: The authors chose to submit the manuscript to PLOS One as there are no strict limits to the length of the article. The authors contend that a study and analysis as complex as the one presented in this study warrants addressing the key pathways known to be impacted by the conditions of the study. We appreciate the reviewer’s comment, however, in the absence of specific feedback, we are unsure which sections to move to supplementary section. 

Comment #5: Another observation is that throughout the manuscript reader has focused to highlight or show the up-regulated genes however the information or description of down-regulated genes has not got due attention.

Response: This is indeed an astute observation, something the authors had noticed during the preparation of the manuscript. The focus on upregulation of genes stems from the specific hypotheses that were evaluated in the context of ORG treatment over CONV. 

Comment #6: Conclusion section is also long, this must not be more than 150-200 words.

Response: As per the instructions to authors, there is no word limit for the conclusion section. The only requirement is that the language should be clear and concise. In the absence of any specific suggestions, the authors feel that the conclusion as written appropriately sums up all the aspects of the study clearly and concisely. 

Comment #7: Probably the raw data of transcriptome has not been uploaded to any publicly available platform such as NCBI or any other, please upload raw data and mention accession number in the manuscript.

Response: In the material and methods section, under the subsection Illumina Sequencing the raw data has been described as being uploaded to the NCBI SRA and the accession numbers were included.

Comment #8: This article may be cited in the discussion as this is closely related with the current study "19. Muhammad Azher Nawaz, Chen Chen, Fareeha Shireen, Zuhua Zheng, Hamza Sohail, Muhammad Afzal, Muhammad Amjad Ali, Bie Zhilong, Yuan Huang. 2018. Genome-wide expression profiling of leaves and root of watermelon in response to low nitrogen. BMC Genomics. 19: 456 " https://bmcgenomics.biomedcentral.com/track/pdf/10.1186/s12864-018-4856-x

Response: Thank you for the suggestion. The published article is pertinent to this work and has been cited in the discussion.

---

## [Decision Letter · Decision Letter 1]

1 Nov 2019

PONE-D-19-25332R1

Concomitant phytonutrient and transcriptome analysis of mature fruit and leaf tissues of tomato (Solanum lycopersicum L. cv. Oregon Spring) grown using organic and conventional fertilizer

PLOS ONE

Dear Professor Dhingra,

Thank you for submitting your manuscript to PLOS ONE. After careful consideration, we feel that it has merit but does not fully meet PLOS ONE’s publication criteria as it currently stands. Therefore, we invite you to submit a revised version of the manuscript that addresses the points raised during the review process.

We would appreciate receiving your revised manuscript by Dec 16 2019 11:59PM. To enhance the reproducibility of your results, we recommend that if applicable you deposit your laboratory protocols in protocols.io, where a protocol can be assigned its own identifier (DOI) such that it can be cited independently in the future. For instructions see: http://journals.plos.org/plosone/s/submission-guidelines#loc-laboratory-protocols

We look forward to receiving your revised manuscript.

Kind regards,

Yuan Huang

Academic Editor

PLOS ONE

Reviewers' comments:

Reviewer's Responses to Questions

**Comments to the Author**

1. If the authors have adequately addressed your comments raised in a previous round of review and you feel that this manuscript is now acceptable for publication, you may indicate that here to bypass the “Comments to the Author” section, enter your conflict of interest statement in the “Confidential to Editor” section, and submit your "Accept" recommendation.

Reviewer #1: All comments have been addressed

2. Is the manuscript technically sound, and do the data support the conclusions?

Reviewer #1: Yes

3. Has the statistical analysis been performed appropriately and rigorously? 

Reviewer #1: Yes

4. Have the authors made all data underlying the findings in their manuscript fully available?

Reviewer #1: Yes

5. Is the manuscript presented in an intelligible fashion and written in standard English?

Reviewer #1: Yes

6. Review Comments to the Author

Reviewer #1: You have not followed the suggestions, and merely changed a few words in the revised version. We understand there is no page limit for PLOSONE articles, but it dose not means authors should write an article of 50 or 70 or 80 pages. Please reduce the length of the document, as for as conclusion section is concerned that should look like a "Conclusion". Reduce it to nearly 150-200 words written in a single paragraph. Please see some previously published articles to understand what does "conclusion" means?

Represent genes expression data in Log2fold change form

7. PLOS authors have the option to publish the peer review history of their article (what does this mean?). If published, this will include your full peer review and any attached files.

Reviewer #1: No

---

## [Author Response · Author response to Decision Letter 1]

7 Dec 2019

December 7, 2019

Dear Editor, 

The authors would like to thank the reviewer for their critical, and very helpful comments. We have incorporated major revisions to reduce the size of the manuscript and have also incorporated the log2fold-change data for gene expression. All the edits are visible in the manuscript with track changes. It is our hope that the manuscript will be acceptable for publication in its revised form. As advised, we have provided a point-by-point response to the reviewers’ comments. The author’s responses to reviewers’ comments are as follows.

Reviewer Comment #1: You have not followed the suggestions, and merely changed a few words in the revised version. We understand there is no page limit for PLOSONE articles, but it dose not means authors should write an article of 50 or 70 or 80 pages. Please reduce the length of the document, as for as conclusion section is concerned that should look like a "Conclusion". Reduce it to nearly 150-200 words written in a single paragraph. Please see some previously published articles to understand what does "conclusion" means?

Author’s Response: The manuscript has been substantially shortened in length. The conclusion section is now 195 words long, and presented as a single paragraph. 

Reviewer Comment #2: Represent genes expression data in Log2fold change form.

Author’s Response: The gene expression data has now been presented in Log2Fold-change form both in the text as well as in the tables accompanying the revised figures in the main manuscript as well as in the supplementary tables and figures.

---

## [Decision Letter · Decision Letter 2]

19 Dec 2019

Concomitant phytonutrient and transcriptome analysis of mature fruit and leaf tissues of tomato (Solanum lycopersicum L. cv. Oregon Spring) grown using organic and conventional fertilizer

PONE-D-19-25332R2

Dear Dr. Dhingra,

We are pleased to inform you that your manuscript has been judged scientifically suitable for publication and will be formally accepted for publication once it complies with all outstanding technical requirements.

With kind regards,

Yuan Huang

Academic Editor

PLOS ONE

Additional Editor Comments (optional):

Reviewers' comments:

Reviewer's Responses to Questions

**Comments to the Author**

1. If the authors have adequately addressed your comments raised in a previous round of review and you feel that this manuscript is now acceptable for publication, you may indicate that here to bypass the “Comments to the Author” section, enter your conflict of interest statement in the “Confidential to Editor” section, and submit your "Accept" recommendation.

Reviewer #1: All comments have been addressed

2. Is the manuscript technically sound, and do the data support the conclusions?

Reviewer #1: Yes

3. Has the statistical analysis been performed appropriately and rigorously? 

Reviewer #1: Yes

4. Have the authors made all data underlying the findings in their manuscript fully available?

Reviewer #1: Yes

5. Is the manuscript presented in an intelligible fashion and written in standard English?

Reviewer #1: Yes

6. Review Comments to the Author

Reviewer #1: (No Response)

7. PLOS authors have the option to publish the peer review history of their article (what does this mean?). If published, this will include your full peer review and any attached files.

Reviewer #1: Yes: Muhammad Azher Nawaz

---

## [Editor Report · Acceptance letter]

23 Dec 2019

PONE-D-19-25332R2 

Concomitant phytonutrient and transcriptome analysis of mature fruit and leaf tissues of tomato (Solanum lycopersicum L. cv. Oregon Spring) grown using organic and conventional fertilizer 

Dear Dr. Dhingra:

I am pleased to inform you that your manuscript has been deemed suitable for publication in PLOS ONE. Congratulations! Your manuscript is now with our production department. 

With kind regards,

on behalf of

Dr. Yuan Huang 

Academic Editor

PLOS ONE